# Breaking barriers: how cytomegaloviruses manipulate myeloid cells to invade tissues

Agnibesh Dey [ID] [1,2,3,4], Vitka Gres[1,2], Alina Nelipovich[1,2,4], Philipp Kolb [ID] [5], Roland Immler [ID] [6], Zsolt Ruzsics[5], Katrin Kierdorf[1,7], Philipp Henneke [ID] [1,2,7,9] & Sebastian Baasch [ID] [1,2,8,9] [✉]

## Abstract

Cytomegaloviruses are highly adapted to their hosts through millions of years of co-evolution. Most human (H)CMV infections occur early in life when the virus enters via the mucosal surfaces of the respiratory or gastrointestinal tract. These infections are usually asymptomatic, but coincide with critical phases of immune development, shaping long-term host immunity. In rare but clinically significant cases, CMV can invade protected sites, such as the central nervous system, leading to symptomatic disease. As the most abundant immune cells at barrier sites, myeloid cells support all stages of the viral life cycle: replication, dissemination, and latency. During the perinatal period, the myeloid compartment undergoes profound changes, including macrophage maturation, monocyte influx, and functional adaptation. CMV may exploit these developmental transitions to establish infection and to cross tissue barriers. This review discusses how CMV manipulates myeloid immune cells to establish postnatal infection, particularly via the respiratory tract, and explores strategies by which CMVs breach the placental barrier and access the fetal brain. The review integrates evidence from multiple CMV species, with emphasis on human and mouse data.

Subject Categories Immunology; Microbiology, Virology & Host Pathogen Interaction; Molecular Biology of Disease

## Introduction

Cytomegaloviruses (CMVs) are strict host-specific β-herpesviruses that usually establish infection early in postnatal life. They have been an integral part of mammalian evolution (McGeoch et al, 2006; Brito et al, 2021; Litvin et al, 2024) and immune ontogeny. As a result of co-evolution, CMVs utilize multiple virus-specific gene products to suppress the host's immune response (Browne and Shenk, 2003; Hengel et al, 1999; Reddehase, 2002; Tomazin et al, 1999; Berry et al, 2020), which benefits both: the virus by facilitating its spread, and the host by avoiding immune-mediated tissue damage (Baasch et al, 2020; Sorci et al, 2013). Nonetheless, immune tolerance comes at a cost, leading to high viral titers, especially in infants, thereby promoting rapid and wide-ranging dissemination (Geris et al, 2022). CMVs typically enter the host via mucosal surfaces, where they infect various cell types and then spread as cell-associated or as cell-free viral particles to distant sites via blood and lymphatic systems. However, the exact mechanisms by which CMVs breach these barriers, and whether multiple or alternative strategies contribute to host entry and spread, remain incompletely understood.

Conceptually, barrier tissues can be divided into outer barriers that primarily protect against infection after contact, such as the gastrointestinal or respiratory tract, and tissue barriers that protect critical sites in the host, such as the fetus and the central nervous system (CNS). It appears that CMVs have particular properties to cross outer barrier tissues and spread through the host without overt symptoms. However, in some settings, for example, in human pregnancy, in premature infants, and in immunodeficiency, these properties then allow the virus to conquer tissue barriers of protected organs, such as the CNS, resulting in severe disease conditions (Griffiths and Reeves, 2021; Boppana et al, 1992).

Although cytomegalovirus (CMV) exhibits broad cellular tropism, cells of the myeloid lineage represent primary targets and important viral reservoirs. From hematopoietic progenitor cells in the bone marrow to circulating monocytes and terminally differentiated tissue macrophages (MACs), all stages of the myeloid lineage contribute to CMV's life cycle: replication, viral dissemination, and latency. Thus, the myeloid lineage is critical for CMV pathogenesis.

Birth represents a profound physiological shift for the organism, including the abrupt transition from a protected intrauterine environment to one that is densely populated by microbes. This transition necessitates rapid adaptation of multiple organ systems, including the immune system (Henneke et al, 2021). For the myeloid lineage in particular, birth initiates a reorganization of cell populations and functional states. Tissue-resident MACs, for

---

[1]Institute for Infection Prevention and Control, University Medical Center, Faculty of Medicine, University of Freiburg, Freiburg, Germany. [2]Center for Chronic Immunodeficiency (CCI), University Medical Center, Faculty of Medicine, University of Freiburg, Freiburg, Germany. [3]Spemann Graduate School of Biology and Medicine, University of Freiburg, Freiburg, Germany. [4]Faculty of Biology, University of Freiburg, Freiburg, Germany. [5]Institute of Virology, University Medical Center, Faculty of Medicine, University of Freiburg, Freiburg, Germany. [6]Institute of Cardiovascular Physiology and Pathophysiology, Biomedical Center, Ludwig-Maximilians-Universität (LMU) München, Planegg-Martinsried, Germany. [7]Centre for Integrative Biological Signalling Studies (CIBSS), University of Freiburg, Freiburg, Germany. [8]Freiburg Institute for Advanced Studies (FRIAS), University of Freiburg, Freiburg, Germany. [9]These authors contributed equally: Philipp Henneke, Sebastian Baasch. [✉]E-mail: sebastian.baasch@uniklinik-freiburg.de

example, populate all organs in humans and mice already during embryogenesis. Postnatal life introduces new signals, such as microbial exposure, oxygen tension, dietary changes, and inflammation, that alter the local tissue milieu and reshape the MAC landscape. For instance, alveolar MACs that derive from embryonic precursors and reside in the air-exposed side of the alveolus in the lung only fully mature within the first week after birth (Guilliams et al, 2013a). In addition, monocytes derived from definitive hematopoiesis in the bone marrow begin to infiltrate various tissues in a tissue-specific and time-dependent manner (Ginhoux and Guilliams, 2016). This process complements or replaces embryonically seeded MACs in certain organs (Mass et al, 2023). In the intestine, MAC turnover increases significantly around weaning (Bain et al, 2014), reflecting adaptation to dietary antigens and commensal microbiota. In the CNS, the perivascular space around the arterioles of the brain only forms in the perinatal phase, instructing resident meningeal MACs to occupy that niche, establishing a new MAC subset (Masuda et al, 2022). During pregnancy, the placenta evolves as a new organ, in which embryonic MACs (Hofbauer cells) are the most abundant fetal immune cells, facilitating the repair of breaches in the placental barrier. Maternal MACs, however, are exclusively derived from monocytes and closely attach to placental cells (Thomas et al, 2021). Accordingly, perinatal changes most likely entail adaptation of MAC core functions with regard to maintenance of tissue integrity, the protection against external threats, and the assistance during tissue repair (Mass et al, 2023). CMV may utilize these perinatal adaptational changes in the myeloid lineage to successfully establish infection of the host.

Hence, understanding the strategies that CMVs use to breach barriers, including activating, infecting and/or manipulating cells in barrier tissues is, both mechanistically and translationally, the key to decipher the establishment of CMV infection in its host (Box 1). This review addresses this topic and further highlights the importance to better understand the differences between perinatal immunity and adult immunity. Since mechanistic studies in humans are mostly restricted to in vitro or ex vivo systems, diverse animal models with species-specific CMVs are needed to better understand the range of CMV-associated pathologies seen in humans. Accordingly, this review integrates data across species, e.g., guinea pig CMV, rat CMV, and non-human primate CMVs. Nonetheless, the discussed literature primarily derives from studies using mouse (M)CMV or human studies using HCMV. To ensure clarity, the CMV species and experimental system will be specified throughout the review, and the main source of the presented data will be indicated at the beginning of each section.

---

**Box 1    Open questions in CMV entry mechanisms and host immune responses**

- What mechanisms enable CMVs to penetrate barrier tissues, such as the respiratory tract, and to initiate infection?
- How do CMVs breach the placental barrier and enter the fetal brain?
- How does perinatal immunity to CMV differ from adult immune responses?
- What role do MACs play in CMV pathogenesis?

---

# The myeloid cell lineage and its nexus with CMV replication, dissemination, and latency

The myeloid lineage arises from different hematopoietic sources during development. Here, tissue-resident MACs have been demonstrated to originate in a stratified manner from different hematopoietic progenitor sources. Starting in early embryogenesis, MACs derived from erythro-myeloid progenitors (EMP) in the yolk sac (at E9.0 in the mouse) and from EMP-derived fetal monocytes of the fetal liver (at E12.5) populate nearly all prenatal tissues at distinct time points during development (Gomez Perdiguero et al, 2015; Yona et al, 2013; Schulz et al, 2012). Most MAC populations are long-lived, remain of prenatal origin, and maintain their entities through endogenous proliferation within their host tissue. After birth, some tissues experience a turnover of embryonic MACs by monocyte-derived MACs, i.e., cells originating from hematopoietic stem cells (HSC) in the bone marrow (Mass et al, 2023; Bain et al, 2014; Molawi et al, 2014; Cugurra et al, 2021). In a simplified model of myeloid ontogeny, HSCs give rise to increasingly lineage-restricted progenitors, such as the common myeloid progenitor (CMP). CMPs further commit to specific cell fates (i.e., granulocyte-monocyte progenitor; GMP or monocyte and dendritic cell progenitor; MDP), which ultimately generate classical monocytes that egress from the bone marrow (Liu et al, 2019; Yáñez et al, 2017; Lösslein et al, 2021). This stepwise myeloid cell differentiation is closely interlinked with CMV infection establishment, dissemination in the host, long-term persistence (latency), and reactivation and will be discussed below.

MACs in barrier tissues are essential to maintain tissue integrity and immune tolerance against commensal microbiota despite continuous physical and biological stresses. At the same time, they are crucial for immune surveillance for incoming infections (Mowat et al, 2017). CMVs encode gene products that target, manipulate, and ultimately overcome control by MACs (Baasch et al, 2021; Daley-Bauer et al, 2017) (Box 1). As an example, Mck2 (m131/m129), as part of MCMV's trimeric entry complex, is essential for MAC infection (Eletreby et al, 2023). Similarly, mutation in the UL128 locus (UL128L) of HCMV's pentameric complex, consisting of the highly conserved glycoprotein (g)H, gL, and UL128L encoding UL128, UL130, and UL131A, impairs infection of leukocytes and especially MACs (Hahn et al, 2004; Cimato et al, 2024). Homologs of the HCMV pentameric complex components were discovered in guinea pig CMV (GP129/131/133) and rhesus macaque CMV (RhUL128/130/131) (Auerbach et al, 2013; Oxford et al, 2008; Lilja and Shenk, 2008). The deletion of the respective homologs results in impaired infection of endothelial or epithelial cells in vitro. However, their role in infecting MACs or attracting myeloid cells has not been tested yet (Auerbach et al, 2013; Lilja and Shenk, 2008). M36, an inhibitor of caspase-8 in all infected host cells, induces a MAC-specific growth phenotype in vitro and is required for MCMV dissemination in vivo (Ebermann et al, 2012). Its HCMV homolog UL36 also inhibits apoptosis in the human monocyte/MAC cell line THP1, and cloning of UL36 into M36-deleted MCMV rescues viral infection kinetics in murine MAC cell line RAW264.7 and in vivo (McCormick et al, 2010; Chaudhry et al, 2017). m140, another example, is required for efficient capsid assembly and thus, viral replication in MACs (Hanson et al, 2009). As a consequence, m140-deficient MCMV shows reduced viral replication in the spleen as

compared to the wild-type and revertant strain after intravenous infection (Hanson et al, 2009). Similarly, m141 facilitates viral replication in MACs, and m141-deficient MCMV shows impaired replication in the spleen, although to a lesser extent than m140-deficient MCMV (Hanson et al, 2001). Another gene product, m139, inhibits type I interferon (IFN I) production specifically in MACs and thus facilitates viral spread, e.g., to the draining lymph node and the salivary gland (SG) as site of latency (Puhach et al, 2020). The HCMV gene US28 and its homolog in MCMV, M33, are constitutively active CMV-encoded G protein-coupled receptors, which signal mainly through $G\alpha_q$/PLC pathways, affecting migration, survival, and inflammatory responses (Davis-Poynter et al, 1997; Casarosa et al, 2001). Infection with M33-deficient MCMV is severely attenuated, especially with regard to SG dissemination and late-phase viral persistence (Bittencourt et al, 2014). Although M33 is expressed across all examined virus-permissive cell types, the reduced dissemination in M33-deficient MCMV is associated with defective spread by infected myeloid cells (Melnychuk et al, 2005; Farrell et al, 2019).

Depending on the route of infection, MACs and dendritic cells (DCs) may both contribute to the initial spread of MCMV, acting as "Trojan horses" (Farrell et al, 2019). Intraperitoneal infection of adult mice causes an immediate viremia with MCMV spreading via $CD11c^+$ DC to the SG. In contrast, MCMV infects $F4/80^+$ MACs to disseminate to the brown adipose tissue adjacent to the SG (Farrell et al, 2019). However, immunophenotypic analyses based on single markers may not suffice to discriminate between CMV-infected myeloid cell types. For example, CD11c is not a specific DC marker across tissues, since peritoneal, intestinal, and alveolar MACs (Bain et al, 2013; Kim et al, 2016; Gautier et al, 2012) also express this integrin subunit. Moreover, MCMV-infection leads to a down-regulation of CD45 and the characteristic MAC markers F4/80, CD64, and CD11c in mice (both in vitro and in vivo) and CD11b in humans (Schwartz et al, 2023; Baasch et al, 2021). Accordingly, labeling of host cells via fate mapping and using fluorescent reporter CMV, or a combination of both, is necessary to reliably characterize and track CMV-infected cells in a spatiotemporal manner (Baasch et al, 2020, 2024; Bošnjak et al, 2023b).

In addition to MACs, monocytes have long been established as CMV targets and putative vehicles for MCMV, HCMV, but also rat CMV dissemination (Collins et al, 1994; Stoddart et al, 1994; Smith et al, 2004a, 2004b; Van Der Strate et al, 2003). However, as opposed to blood, spleen, and bone marrow, monocytes are relatively rare in other resting tissues compared to MACs. Upon recruitment to tissues, classical monocytes differentiate into permissive MACs, thereby losing their monocyte identity (Bain et al, 2014; Molawi et al, 2014; Liu et al, 2019; Kolter et al, 2019; Shi and Pamer, 2011). Accordingly, classical monocytes rather play a role during latency and sporadic reactivation (as described below). Circulating non-classical monocytes, however, are less recognized to enter the tissue and differentiate into MAC. They patrol the endothelium and could get infected via endothelial cells at the initial entry site of MCMV or HCMV. Once infected, these monocytes continue to circulate and contribute to viral dissemination throughout the body (Baasch et al, 2020; Daley-Bauer et al, 2014). Both subsets of monocytes therefore rather act as secondary targets, becoming infected after CMV has breached barrier tissues and undergone initial replication cycles. During this phase, MCMV-induced host chemokines such as CCL2, along with the viral CC chemokine homolog Mck2, drive monocyte recruitment from the bloodstream to primary sites of infection (Saederup et al, 1999; Daley-Bauer et al, 2014).

HSCs are considered a site for HCMV latency in the bone marrow (Mendelson et al, 1996; Hahn et al, 1998; Goodrum et al, 2002). Interestingly, CMV encodes gene products that drive myeloid lineage commitment and monocyte differentiation at the progenitor stage. The HCMV glycoprotein pUL7 binds to the Fms-like tyrosine kinase 3 receptor (FLT3R) and increases myeloid colony formation from HSC ex vivo (Crawford et al, 2018). Accordingly, blood monocytes were found to be increased in humanized NOD/scid-IL2Rγ mice as late as 8 weeks after infection (Crawford et al, 2018). GMPs, as committed progenitor cells of the myeloid lineage, have also been found to serve as a viral reservoir and transmit HCMV to cells that are susceptible to lytic infection (Kondo et al, 1994; Zhuravskaya et al, 1997). These progenitors mature into monocytes that harbor HCMV without active replication (Taylor-Wiedeman et al, 1991; Shnayder et al, 2020). Upon differentiation of infected monocytes into MACs, HCMV reactivates replication ex vivo (Söderberg-Nauclér et al, 1997, 2001; Minton et al, 1994). Notably, HSCs, GMPs, and monocytes do not seem to be the only myeloid cells accounting for HCMV persistence. Non-productively infected MACs can reactivate infectious viral progeny ex vivo (Schwartz et al, 2023). Additionally, data from patients that underwent solid organ or stem cell transplantation, as well as data from mouse models, suggest that tissues beyond the bone marrow niche might serve as a site of latency, including MACs in the lung and fibroblasts (Schwartz and Stern-Ginossar, 2023; Balthesen et al, 1993; Sitnik et al, 2023; Pollock et al, 1997; Koffron et al, 1998).

Overall, cells of the myeloid lineage and MACs in particular support the entire life cycle of CMV, from being the initial target, serving as a vehicle for dissemination, to being putative sites of CMV latency and reactivation. These properties are in line with the observation that CMV dedicates a plethora of gene products to specifically target MAC programs (Baasch et al, 2020). Yet, as outlined above, CMV causes a detrimental subversion of the immunophenotype of MCMV- and HCMV-infected MACs, including impairment of key functions, such as phagocytosis or cytokine release in response to bacteria, and changes in surface marker expression (Baasch et al, 2021; Schwartz et al, 2023). Thus, the role of MACs may have been underestimated. Future studies using novel high-throughput cell profiling approaches in combination with cell and virus fate mapping across the course of infection are required to dissect the changes of the immunophenotype and function in the myeloid compartment by CMVs.

## Breaching barriers to establish postnatal infection

After birth, HCMV is transmitted via body fluids, such as saliva, urine and breast milk (Dworsky et al, 1983; Cannon et al, 2014), however, the precise primary site for host entry is not clear yet. Overall, the digestive and respiratory tracts, as the largest mucosal surfaces of the human body, are considered the primary CMV entry sites. Although virus-containing breast milk is swallowed by infants, HCMV may reach the respiratory tract via aerosol formation or micro-aspiration during breastfeeding. Ultimately,

CMV needs to overcome protective layers of mucus in the intestine or surfactant in the lung to cross the epithelial layer and to escape the control of resident immune cells in order to get access to the tissue.

Importantly, and as outlined before, both the intestine and lung undergo rapid and tissue-specific remodeling of the myeloid compartment during the perinatal period. In the intestine, weaning triggers monocyte influx and replacement of embryonic MACs in the lamina propria (Bain et al, 2014). In the lung, alveolar MACs require the first week of life to fully mature (Guilliams et al, 2013a). CMV infection during these transitional phases is likely to influence both the course of infection and the developing immune response. These dynamic shifts in the local immune landscape may also present windows of opportunity that CMVs can exploit to establish infection.

In this section, we explore how different myeloid cells contribute to CMV infection and how they are exploited by MCMV to cross mucosal barriers.

## The lung

In the respiratory tract, a single layer of alveolar epithelial cells (AECs) connected via tight junctions forms a physical barrier for particles and pathogens. AECs comprise AEC type 1, which cover around 97.5%, and AEC type 2, which cover around 2.5% of the alveolar surface in rats, with a linear increase in the surface area-to-cell number ratio in mammals (Haies et al, 1981; Stone et al, 1992). AECs type 1 form a thin layer, enabling rapid gas exchange, while AECs type 2 produce surfactant, which reduces surface tension at the air-liquid interface and exhibits opsonic and immunomodulatory functions (Guillot et al, 2013).

Following experimental intranasal inoculation, MCMV infects the lung and spreads to mediastinal lymph nodes and the SG (Oduro et al, 2016). Two major myeloid cell types in the lower respiratory tract have been proposed to carry MCMV across the mucosal lung barrier: alveolar MACs and lung DCs. Both cell types express CD11c, which hampers their discrimination based on this single marker. To date, it is not entirely clear which cell type (or whether both) spreads MCMV from the lung to distant sites (Box 1). Thus, we follow the nomenclature according to the referenced source.

Alveolar MACs, which on average patrol three alveoli, are essential for early pathogen resistance, initiation of the immune response, clearance of cellular debris, and surfactant metabolism (Westphalen et al, 2014; Hussell and Bell, 2014). They can be identified in the mouse lung by a combination of surface markers (CD11c$^+$, SiglecF$^+$, CD11b$^-$) and high autofluorescence. However, during infection, they may upregulate CD11b expression or downregulate other surface markers depending on their infectious status (Baasch et al, 2021, 2024).

Conventional DC (cDC) can be divided into two distinct subsets: cDC1 are characterized by the expression of the E-cadherin-binding integrin CD103 (*Itgae*), the lymphotactin receptor XCR1 and DNGR-1 (*Clec9a*), while lacking CD11b (*Itgam*) expression. In the lung, they are located in close proximity to the epithelial layer, especially of the conducting airways, and are thus well positioned to rapidly access inhaled antigens and initiate an antiviral CD8$^+$ T-cell response (Guilliams et al, 2013b). In contrast, cDC2 are CD11b$^+$ and SIRPα$^+$, while negative for CD103, XCR1 or DNGR-1. They are pivotal to induce T helper type 2 response, but can thereby also contribute to asthma (Hammad and Lambrecht, 2021).

Despite the overall broad cell tropism of CMVs, alveolar MACs are the predominant target of the virus in acute intranasal MCMV infection (Baasch et al, 2021; Farrell et al, 2017, 2016a). Host-pathogen fate mapping, in which the host encodes a cell type-specific cre-recombinase (*Itgax*-cre), while the pathogen harbors a STOP-floxed GFP cassette (MCMV$^{LSL-GFP}$) was used to trace MCMV-infected cells in vivo (Baasch et al, 2021, 2024). This approach revealed that GFP$^+$ cells expressed alveolar MAC markers, such as CD64 and SiglecF, 12 hours post infection, defining alveolar MAC as initial targets of MCMV (Baasch et al, 2021). Notably, within 3 days after infection, this surface marker expression was globally downregulated, highlighting again the difficulty in immunophenotyping infected cells over the course of infection, especially following several viral replication cycles (Baasch et al, 2021).

Although alveolar MACs are the primary target following CMV infection of the respiratory tract, their role in initial virus propagation and severity of infection is still under debate. On the one hand, depletion of lung MACs in adult $Cd169^{DTR}$ mice substantially increased MCMV lung titers 1 day post infection (Farrell et al, 2016a), indicating that alveolar MACs mitigate the infection (Baasch et al, 2020). On the other hand, clodronate liposome-mediated depletion of alveolar MACs yielded in decreasing MCMV levels in the lung after infection of neonatal mice (Stahl et al, 2015). The differences in infection outcomes might depend on the applied depletion model, i.e., pharmaceutical or genetic, with varying deletion kinetics and efficiencies. However, it could very well be suggestive of the opposing roles of adult versus neonatal MACs in viral propagation (Box 1). Interestingly, depletion of alveolar MACs not only leads to less virus in the lung, but also to reduced viral dissemination to the liver or brain in neonatal mice (Stahl et al, 2015). Infected alveolar MACs that were adoptively transferred from $Itgax^{cre/+}$:ROSA26$^{LSL-Tomato}$ into $Csf2rb^{-/-}$ mice translocate to the lung interstitium and migrate to the mediastinal lymph node after infection of adult mice (Baasch et al, 2021). These findings indicate a shared MCMV strategy that exploits the reprogramming of alveolar MACs into invasive, migratory cells to enable dissemination independent of age. Importantly, upon other inflammatory stimuli, such as LPS, alveolar MAC do not travel to the lymph node (Cleary et al, 2025). The transcription factor *Zeb1* controls the migration of MCMV-infected alveolar MACs to the mediastinal lymph node (Fig. 1) as demonstrated in myeloid-specific knock-out mice ($LysM^{cre/+}$:$Zeb1^{floxed}$) (Baasch et al, 2021). Notably, only a minority of infected alveolar MACs migrate to the interstitium and to the draining lymph node, indicating that a combination of modulating factors beyond *Zeb1* underlies this peculiar alveolar MAC transformation.

Alongside alveolar MACs, DCs have also been reported to act as a potential "Trojan horse" for viral dissemination throughout the host. It was reported that DCs mediate systemic viral spread to the SG after intranasal infection as well (Farrell et al, 2017). In this study, DCs and "DC-like alveolar MACs" were distinguished from CD11c$^+$ alveolar MACs through poor uptake of the dye PKH26 before and after infection (Farrell et al, 2017). However, since MCMV impairs phagocytic activity in MACs (Baasch et al, 2021; Pesanti and Shanley, 1984), cellular labeling that relies on phagocytosis may be error-prone. Moreover, infected CD11c$^+$

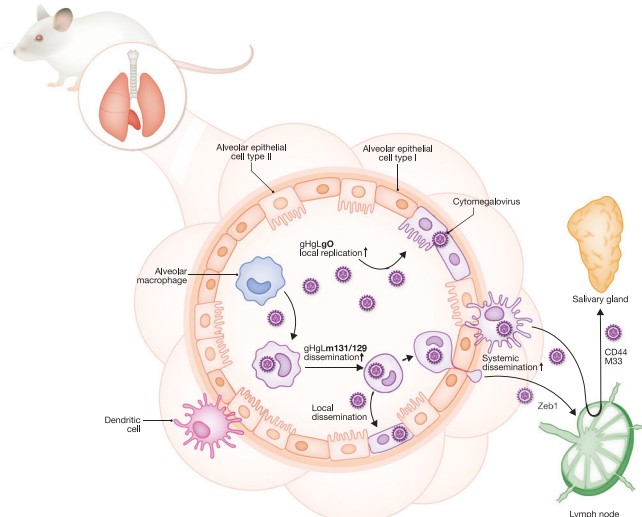

**Figure 1. MAC and DC in the lung facilitate local and systemic dissemination.**

In the lung, MCMV targets cells, such as alveolar MACs that reside directly in the alveoli and cDC1 adjacent to the epithelium. After viral m131/129 (Mck2)-dependent infection of alveolar MACs, they spread CMV locally to CD11c⁻ epithelial cells (purple AEC) and/or migrate to the interstitium and further to the draining mediastinal lymph nodes in a *Zeb1*-dependent manner. Dissemination of infected cDCs from the lymph node to the salivary gland requires expression of CD44 and M33. AECs can also be directly infected by CMV, which is dependent on the viral gO.

DCs spreading the virus to the SG were defined as being negative for the murine MAC marker F4/80 (Farrell et al, 2019), which is efficiently downregulated in bone marrow-derived MACs already 16 h post MCMV infection in vitro (Baasch et al, 2021). Thus, the turnover and degradation of surface proteins in infected cells may differ between proteins, cell types, and tissues, which may explain the "punctate" detection of CD11c in microscopy on infected cells in mediastinal lymph nodes (Farrell et al, 2017; Yunis et al, 2019).

Mechanistically, migration of infected CD11c⁺ DCs from the lung into lymph nodes and further to the SG is facilitated by the surface molecule CD44 and the MCMV-encoded chemokine receptor homolog M33 (Fig. 1) that causes constitutive $G\alpha_q$-protein-coupled signaling (Farrell et al, 2019, 2017). As described above (see the section "The myeloid cell lineage and its nexus with CMV replication, dissemination and latency"), infection of MACs, including alveolar MACs, critically depends on the trimeric gH, gL, and Mck2 cell-entry complex (Fig. 1). This process relies on expression of MHC class I/B2m on the MAC surface (Stahl et al, 2015; Bošnjak et al, 2023a). Mck2 mediates early alveolar MAC infection, thereby attenuating viral replication (Farrell et al, 2016a), and enabling viral spread by transformed alveolar MACs (Fig. 1) (Baasch et al, 2021). Lack of viral Mck2, and thus reduced alveolar MAC infection, increases viral titers in the lung 2 days after infection (Farrell et al, 2016a). In contrast, lack of gO, which mediates infection of AECs (Fig. 1), dramatically reduces MCMV lung titers after infection (Yunis et al, 2019). Taken together, the absence of Mck2 promotes infection of AECs and thereby enhances local viral replication, at the expense of efficient dissemination to the SG via CD11c⁺ cells (Farrell et al, 2016a). In the presence of Mck2, however, infection of alveolar MACs attenuates viral

replication (Farrell et al, 2016a), spreading the virus to CD11c⁻ cells (Fig. 1) (Baasch et al, 2021). Nonetheless, the differences in viral titers between infection with WT or Mck2-deficient MCMV disappear during the course of acute infection (Bošnjak et al, 2023a). Interestingly and in contrast to the situation in adult mice, Mck2-deficient MCMV is impaired in infecting the respiratory tract as effectively as the Mck2-restored MCMV in a neonatal infection model (Stahl et al, 2015), again suggesting opposing age-dependent consequences of early alveolar MAC infection with MCMV.

The lung also represents a potential site of CMV latency (Balthesen et al, 1993). CD11b⁺ MACs (Koffron et al, 1998) and pulmonary fibroblasts (Sitnik et al, 2023) were identified to bear viral DNA 5-6 months after intraperitoneal or intranasal infection with MCMV, respectively. Interestingly, in hematopoietic stem cell transplantation, the serostatus of the graft recipient represents a greater risk factor for respiratory complications associated with HCMV reactivation, i.e., pneumonia, than that of the donor (Travi and Pergam, 2014). This observation underlines the potential impact of resident cells, such as alveolar MACs, in the lung as sites of CMV latency and reactivation (Poole et al, 2015).

Despite significant progress in understanding CMV infection of the respiratory tract, the distinct contributions of DCs and MACs in viral entry, dissemination, and immune modulation remain under debate. This ambiguity is compounded by overlapping marker expression and dynamic surface changes upon infection. Dissecting the precise roles of these myeloid cell types via cell-specific lineage tracing models paired with viral reporters is critical for understanding which cell CMV utilizes to breach barrier tissues and disseminates systemically after primary postnatal infection of the lung. Importantly, there is emerging evidence that initial neonatal CMV infection differs from adult CMV infection with regard to the role of alveolar MACs (Box 1). This may result in differences in viral control, dissemination, and establishment of latency. Consequently, it could influence the susceptibility to future secondary infections following neonatal infection with CMV as compared to CMV infection acquired later in life. Therefore, age must be considered a critical variable when studying CMV pathogenesis. Ideally, experimental models should include parallel infections in neonatal and adult mice to directly compare immune responses and viral dynamics across developmental stages. Understanding the mechanisms by which alveolar MACs either suppress or enhance viral replication holds significant translational relevance.

## The intestine

The intestinal barrier is composed of a single epithelial layer (enterocytes), firmly connected through protein networks, including tight junctions. Additionally, specialized epithelial cells fulfill particular functions, e.g., goblet cells produce the luminal mucus throughout the intestine, while Paneth cells in the small intestine secrete antimicrobial peptides (Ayabe et al, 2000).

Colonization of the infant with microorganisms occurs immediately after birth, primarily by maternal microbiota, including those in the breast milk (Pannaraj et al, 2017). Similarly, HCMV may reactivate in seropositive mothers and is shed into human milk (Hayes et al, 1972), enabling transmission to the infant (Dworsky et al, 1983; Hamprecht et al, 2001). Thus, the gastrointestinal route is a probable site for HCMV infection in breast-fed infants. MCMV infection of mouse pups via milk has also been reported (Wu et al, 2011), yet

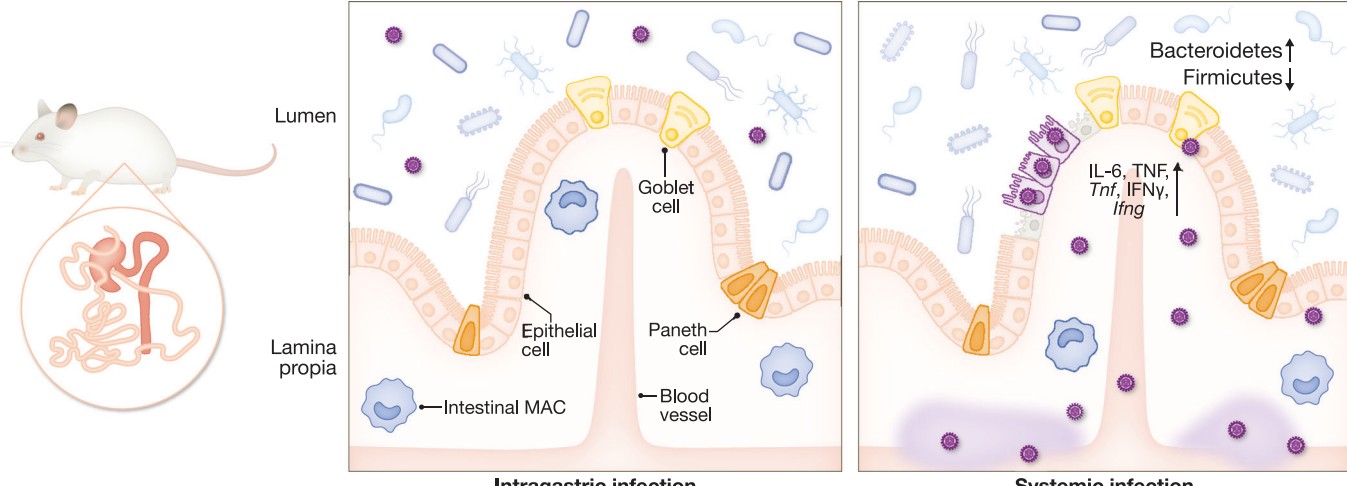

**Figure 2. Systemic MCMV infection causes intestinal inflammation and epithelial cell death.**

In the intestine, experimental intragastric infection (left) does not result in efficient MCMV infection. In contrast, systemic infection (right) leads to viral replication in cells of unknown identity within the lamina propria and muscularis, as well as in epithelial cells, leading to epithelial cell death. This is accompanied by increased expression of *Tnf* and *Ifng*, elevated production of TNF, IFN-γ, and IL-6, and alterations in the intestinal microbiota composition. *For visualization purposes, data from the ileum and colon were combined.*

intragastric infection appears to be inefficient in adult mice (Oduro et al, 2016). In some cases, oral administration of rhesus CMV leads to infection and subsequent horizontal transmission, especially in infant rhesus macaques (dela Pena et al, 2012). However, direct comparison of nasal and oral MCMV infection in neonatal mice demonstrated that successful infection occurs exclusively via the respiratory tract (Farrell et al, 2016b). Moreover, breast-fed pups from latently infected dams showed the highest viral transmission to the lungs (Wu et al, 2011), indicating that the respiratory tract is the dominant site of infection, even when viral exposure occurs orally or through contaminated milk. HCMV is frequently detected in colon-associated diseases in immunocompromised adult patients suffering from inflammatory bowel disease, like Crohn's disease or particularly in ulcerative colitis, and is sporadically found in elderly patients with comorbidities (Rafailidis et al, 2008; Fakhreddine et al, 2019). While its exact role remains debated, HCMV is more likely to exacerbate existing disease rather than initiate it. This observation is associated with an inflammatory phenotype driven by HCMV-infected monocytes that differentiate into intestinal MACs, via Smad7 upregulation and enhanced nuclear translocation of NFκB (Dennis et al, 2018). In neonatal mice, intraperitoneal infection with MCMV causes infectious foci in the lamina propria/muscularis and triggers an inflammatory response indicated by higher *Ifng* and Tumor necrosis factor (*Tnf*) gene expression in the ileum. However, it is unable to initiate necrotic enterocolitis alone (Fig. 2). Inducing necrotic enterocolitis in mice that were co-infected systemically with MCMV leads to exaggeration of the disease in an enterocyte-specific TLR4-dependent manner (Scheese et al, 2025). After systemic infection, MCMV reaches the epithelial cell layer, induces apoptosis, and disrupts intestinal barrier function in the colon of adult mice (Le-Trilling et al, 2023). This infection is accompanied by elevated production of TNF, IFN-γ, and interleukin (IL)-6, as well as alterations in the intestinal microbiota (Le-Trilling et al, 2023). Infection of mouse enteroids confirmed susceptibility of epithelial cells (Le-Trilling et al, 2023) and TLR4 upregulation (Scheese

et al, 2025). In HIV patients, HCMV infects epithelial cells and compromises barrier integrity partially via IL-6 induction, similar to the mouse model (Maidji et al, 2017). Rhesus CMV also leads to a change in the microbiome composition of the gut in naturally or experimentally infected non-human primates (Chin et al, 2022; Santos Rocha et al, 2018). Additionally, Simian immunodeficiency virus-infected non-human primates, as a model to study HIV infection in humans, showed CMV-associated masses in the intestinal tract, including infected MACs (Hutto et al, 2004).

Together, there is currently no clear evidence supporting the intestine as a primary site of CMV infection, suggesting that CMV does not breach the intestinal epithelial barrier to establish an infection. CMV infections may lead to changes of the microbiome composition with consequences for the immune system. However, intestinal pathology associated with CMV appears to result only under immunosuppressive conditions or intestinal comorbidities. Nonetheless, the intestine might represent a prominent site for reactivation due to the continuous replenishment of tissue MACs through monocyte-derived MACs, even under steady state conditions. In settings of immunosuppression, CMV-infected monocytes may differentiate into tissue MACs, triggering viral reactivation and enabling uncontrolled viral replication due to impaired immune control, which can ultimately exacerbate underlying intestinal disease.

# Breaching barriers to establish prenatal infection

## The placenta

The placenta is a unique organ, formed by embryonic/fetal cells to connect the bloodstreams of two genetically distinct individuals. It is essential that the placenta allows the selective exchange of

molecules with the growing fetus, while limiting the transfer of other macromolecules, cells, and pathogens. While these over-arching goals are identical in all eutheria, there are significant differences in placentation and pregnancy between humans and various animals that are often used as model organisms (Carter, 2020). Mice, for example, differ greatly from humans in the cell structure that separates maternal and fetal blood (Ander et al, 2019). Although both species have a hemochorial placenta without maternal epithelial or endothelial cells at the intersection of the two bloodstreams, mouse fetal capillaries are surrounded by three layers of trophoblast cells, whereas in the human placenta, the blood flows directly into the intervillous space, which is lined only with syncytiotrophoblasts (STBs) (Carter, 2020). In addition, mice give birth to so-called altricial pups, in which much of the organ development occurs after birth and not during pregnancy (Carter, 2020). However, scientists studying CMV encephalitis are using the latter to their advantage by infecting neonatal pups at postnatal day 1 to model CNS pathologies resulting from congenital HCMV infection in humans (see the section "The central nervous system"). MCMV infection in pregnant mice leads to resorption of the embryo and stillborn fetuses (Johnson, 1969), similarly to the situation in humans (~15% with primary CMV infection during early pregnancy) (Griffiths and Baboonian, 1984). In contrast to HCMV, however, MCMV appears unable to cross the mouse placental barrier. This is likely related to the aforementioned differences in the structure of the placenta. Guinea pigs, on the other hand, serve as a valuable model because their placenta is monolayered, similar to that of humans, and congenital CMV infection occurs naturally in this species (Carter, 2020; Schleiss and McVoy, 2010). However, relatively little is known about guinea pig CMV at the genetic and molecular levels, which hinders progress in elucidating the mechanisms of viral transmission. Significant progress has been made in developing the non-human primate model (Manuel et al, 2025), particularly using CD4-depleted pregnant dams, which enhances the rate of congenital infection with non-human primate CMV (Bialas et al, 2015). Interestingly, despite the reduced dissemination in pregnant dams in vivo, rhesus CMV induced transplacental fetal infection regardless of the pentameric complex (Wang et al, 2025). However, a functional role of rhesus CMV's pentameric complex with respect to myeloid cells has not formally been shown (see the section "The myeloid cell lineage and its nexus with CMV replication, dissemination and latency"). Human placental organoid models have provided valuable insights into which cell types are susceptible to infection. These studies suggest that infection in cytotrophoblasts appears to be non-productive and show that STB might lack permissiveness due to the absence of *Nrp2* expression (Yang et al, 2024, 2022; Hatterschide et al, 2025). However, most of our current under-standing still relies on limited experimental data and clinical observations in humans. In the following section, we will mainly discuss human data to elaborate on how HCMV can cross the placental barrier.

Throughout gestation, the placenta undergoes extensive restruc-turing, forming a large contact surface between the maternal and fetal circulation, almost entirely enveloped by STB, leaving virtually no intercellular spaces (Aplin, 2010; Robbins et al, 2010). At the same time, physical breakdown of the trophoblast layer has been observed as a physiological process in the growing placenta (Burton and Watson, 1997). Despite the advantage of lacking intercellular

space in a syncytial barrier, such a focal defect would immediately expose its underlying structures. Although repair mechanisms have been identified (Burton and Watson, 1997), cell-free CMV or cell-associated CMV could seize this opportunity to translocate into the placenta and target, for example, fetal Hofbauer cells (Fig. 3). Indeed, Hofbauer cells have been shown to carry HCMV antigen in samples of HCMV-associated placentitis (Sinzger et al, 1993). Hofbauer cells are a heterogeneous population of MACs represent-ing the most abundant fetal immune cell in the human placenta. They reside in the stroma of fetal villi and have microbiocidal capacity in case of syncytial layer disruption (Thomas et al, 2021).

It has been suggested that CMV exploits MACs even for initial transmission from mother to placenta and eventually to the fetus, albeit without experimental evidence (Delorme-Axford et al, 2014). In the decidua, maternal MACs represent the second largest leukocyte population (Sureshchandra et al, 2023; Vento-Tormo et al, 2018), following a unique natural killer (NK) cell population (Koopman et al, 2003). These placenta-associated maternal MACs derive from peripheral blood monocytes and closely attach to the STB layer of the placenta, aiding its repair (Thomas et al, 2021). Similarly to the breach of the lung barrier (see the section "The lung") (Baasch et al, 2021), CMV-infected maternal MAC could provide a possible gateway into the fetal placenta and the developing fetus (Fig. 3). Given their origin, one could speculate whether infected monocytes from the periphery differentiate into maternal MACs or whether maternal MACs are directly infected at the placental interface (Fig. 3).

In addition to the proposed involvement of infected maternal MACs in facilitating transplacental transmission, alternative cell-independent pathways of viral entry into the placenta have been suggested. Antibodies, in particular IgG, are a component of the maternal immunity that tolerates foreign human tissue and can be transferred to the fetus. Starting around week 20 of gestation, maternal IgG is transported across the STBs by the neonatal Fc-receptor (FcRn) to build up a so-called "maternal passive immunity" that mediates immune protection for the developing human fetus (Borghi et al, 2020; Simister, 2003; Simister et al, 1996). Thus, a logical strategy to target congenital HCMV infection is the administration of hyperimmunoglobulin (HIG) preparations of high neutralizing potential. However, HIG treatment during pregnancy has seen varying degrees of success (Hughes et al, 2021; Kagan et al, 2021; Revello et al, 2014; Blázquez-Gamero et al, 2019; Nigro et al, 2005). The need for high doses of HIG suggests that HCMV has evolved mechanisms to evade IgG-mediated immunity. Using a non-human primate model of CMV infection demon-strated that rhesus CMV actively circumvents IgG-mediated immunity by deploying a set of IgG-binding glycoproteins called viral Fc-gamma receptors (Otero et al, 2025). Similar proteins have been identified in HCMV through previous in vitro studies, indicating that HCMV likely employs the same strategy (Corrales-Aguilar et al, 2014; Kolb et al, 2021). This immune evasion may contribute to increased viral load, thereby raising the risk of congenital infection. Moreover, free virions may even exploit the presence of IgG during the earliest stages of infection. Indeed, there is an evidence supporting IgG-dependent transcytosis via FcRn across the STB layer, where HCMV virions enter underlying cytotrophoblasts and are subsequently taken up by fetal Hofbauer cells, ex vivo (Maidji et al, 2006) (Fig. 3A). While it is unclear why HCMV would differ from other pathogens in this mechanism, it

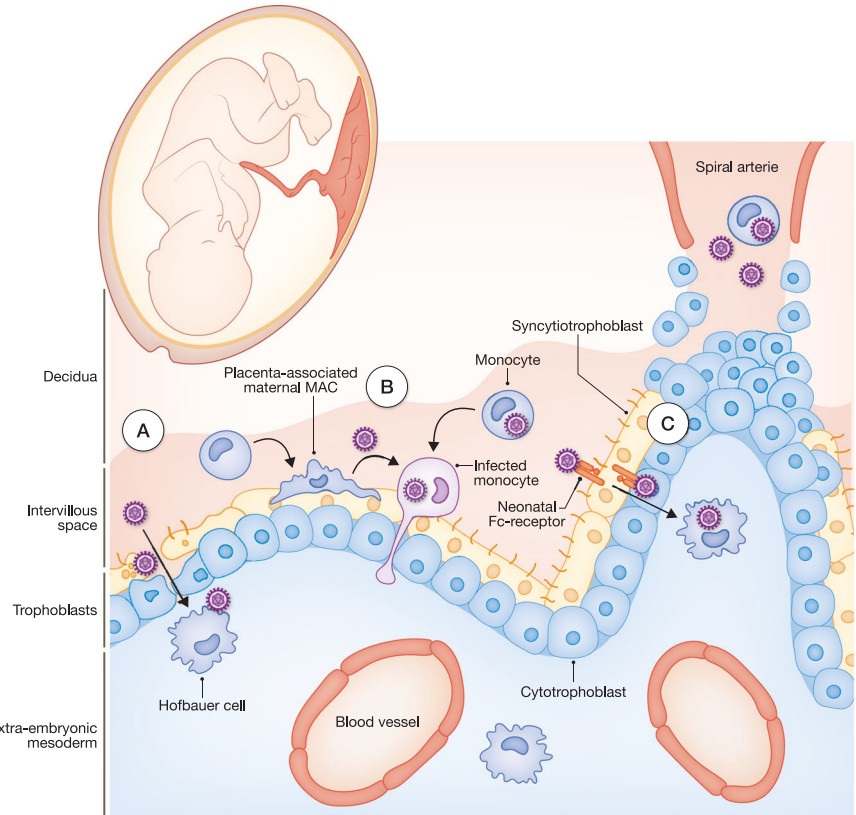

**Figure 3. Routes of CMV entry at the maternal-fetal interface.**

The trophectoderm that consists of trophoblasts surrounds the inner cell mass and is thereby essential for the attachment to the endometrium and the implantation into the uterus wall. Postmitotic trophoblast cells later fuse to produce multinucleated STBs as a part of the chorion. STBs are fed from an underlying layer of cytotrophoblasts, which may also fuse to maintain and expand the syncytium. At the contact site with the decidualized endometrium (Decidua), the STBs disappear, and a subset of CTBs invade the uterine wall as extra-villous trophoblasts to anchor the placenta deep in the maternal decidua (anchoring villus) and remodel the uterine spiral arteries. Subsequently, maternal blood is flowing into the intervillous space. This enables direct contact of CMV (cell-free virion or infected monocyte) with the placenta. Other villi that do not attach to the evolving decidua float in the intervillous space (floating villus) and are completely delimited by STB. (A) Damage of the STB and/or CTB layer enables a free passage of CMV to the placenta, resident Hofbauer cells, and eventually fetal circulation. (B) CMV-infected maternal MACs that are closely attached to the STB layer are transformed and migrate to the placenta. (C) Antibody "covered" CMV in the intervillous space can be transcytosed via neonatal Fc-receptors through STBs to ultimately reach Hofbauer cells.

can be speculated that a virus notoriously resistant to neutralization yet capable of binding IgG (Falk et al, 2018; Kolb et al, 2021) might actually benefit from being coated with pre-existing non-neutralizing antibodies and subsequently transported across the barrier.

In summary, the mechanisms by which HCMV breaches the placental barrier are complex and might vary depending on the gestational stage (Box 1). To date, several mechanisms have been investigated to untangle CMV's strategy. Damage of the STB layer, antibody-mediated transcytosis of CMV through STBs, followed by infection of Hofbauer cells (Fig. 3A,C) or induced migration of infected maternal MACs towards the fetal placenta may be involved (Fig. 3B). Development of new animal models or better controllable in vitro/ex vivo experiments, such as organoid culture derived from primary cells (Yang et al, 2022) that allow cell type-specific tracing, are needed to better dissect mechanisms of viral invasion. Such approaches will help to define the contribution of placental stromal cells and myeloid immune cells to viral invasion and to determine the possible pivotal role of placenta-associated maternal MACs or Hofbauer cells during HCMV infection.

## The central nervous system

Primary maternal HCMV infection, as well as reactivation during pregnancy, can lead to congenital CMV infection (Enders et al, 2011; Revello, 2004). Once CMV has overcome the placental barrier, it can be transmitted to the fetus and disseminate to multiple organs such as the liver, spleen, and importantly, the CNS. The severity of fetal HCMV infection is closely linked to the timing of the infection: fetuses infected during the first two trimesters are at greater risk of developing severe complications, such as encephalitis, hearing loss, neurological impairments, developmental delays, and ocular issues associated with CNS involvement (Pass et al, 2006; Handley et al, 2021). To investigate transplacental transmission of CMV, guinea pigs and non-human primates such as rhesus macaques are utilized (see the section "The placenta"). In guinea pigs, acute infection of dams with guinea pig CMV leads to infection of the SG and, in some cases, the brain of the offspring (Griffith et al, 1982). CMV-induced brain lesions in guinea pigs closely resembled those observed in human infants congenitally

infected with HCMV (Griffith et al, 1982; Hanshaw, 1971). Similarly, intravenous infection of pregnant rhesus macaques led to the infection of various fetal brain regions, although in only 1 out of 12 fetuses (Manuel et al, 2025). Furthermore, to better model human neurodevelopment, human induced pluripotent stem cells-originated cerebral organoids are employed and resemble the malformation of the fetal brain in the context of HCMV infections (Brown et al, 2019; Egilmezer et al, 2024). The pathogenesis of congenital CMV infection of the brain has been widely studied (Krstanović et al, 2021); however, the specific mechanisms through which CMVs breach the barriers of the CNS remain poorly understood. Interestingly, while early postnatal infection of mice results in CNS invasion by MCMV, the ability to infect the brain ceases after postnatal day 12 (Krstanović et al, 2024), raising the question which processes of early postnatal CNS development are associated with CMV actively infecting the brain. Immune and non-immune cells in the CNS interfaces populate and mature pre-, but also postnatally, in a defined spatiotemporal pattern. In the following sections, we will go through the different layers and barriers of the CNS interface and speculate on the involvement of MACs and structural cells in the breach of the CNS. Given the overall limited number of studies examining CNS barriers in the context of CMV infection, this section primarily focuses on data derived from mouse models, which have yielded valuable insights into the complex development and maturation of barrier structures and associated myeloid immune cells.

## CNS barriers

The brain, as a protected organ, is guarded by a multi-layered immune system largely located in the CNS interfaces surrounding the CNS parenchyma, including the meninges, the perivascular space along the arteries, and the ventricular system. Whereas the CNS parenchyma is occupied by a population of tissue-resident MACs, the microglia, the CNS interfaces harbor a very complex and dynamic immune system. It is formed by both structural and immune cells that create interlinked, flexible barriers (Amann et al, 2023; Kierdorf et al, 2019). The meningeal barrier is primarily formed by complex structures of highly specialized collagenous "connective" tissue membranes enclosing the brain and the spinal cord. Traditionally, meninges were divided into three tissue layers: the dura mater attached to the skull, followed by the arachnoid and pia mater, which are sometimes jointly referred to as leptomeninges (Patel and Kirmi, 2009). The dura mater, with an outer periosteal and inner meningeal layer, contains lymphatic vessels (Absinta et al, 2017; Ahn et al, 2019), fenestrated blood vessels, sensory nerves, and a plethora of resident and infiltrating immune cells. In the leptomeninges, the arachnoid mater segregates the interstitial fluid from the dura mater and the cerebrospinal fluid (CSF) from the subarachnoid space. The underlying pia mater is in very tight contact with the CNS parenchyma (Alves De Lima et al, 2020; Rua and McGavern, 2018). Pia mater also covers most blood vessels in the subarachnoid space, which project into the brain and branch out to form small capillaries. The multi-layered compartment that follows the pia mater and segregates the CNS from the bloodstream, including the capillaries, is called the blood-brain barrier (BBB). BBB allows for the selective diffusion of nutrients and essential macromolecules and restricts access of toxic substances, immune cells, and pathogens to the parenchyma (Abbott et al, 2010).

Maturation of the BBB spans the perinatal period and may enable CMV transmission to the CNS during that time. Vascular

endothelial cells, as a pivotal part of the functional BBB, are interconnected by tight junctions, followed by a layer of smooth muscle cells (around arteries and arterioles) or pericytes (around capillaries) and a basement membrane. Astrocytes and their endfeet form a second basement membrane adjacent to the endothelial basement membrane to further limit access to the CNS parenchyma. The formation of the BBB is a multi-step process that begins with sprouting angiogenesis directed by the neuroepithelium during late organogenesis (Risau, 1997; Abbott et al, 2010). Neural progenitors secrete factors that induce endothelial cell migration into the CNS. At the same time, endothelial cells secrete signals to recruit pericytes (Hellström et al, 1999). Thus, vascularization already occurs prior to the generation of astrocytes, which are a critical constituent for maintaining barrier functions of the mature BBB. However, during their invasion, endothelial cells start to form multiple tight junctions (Daneman et al, 2009). Therefore, already at E15 the complexity of tight junction components restricts the permeability of the mouse brain (Ben-Zvi et al, 2014). This argues against BBB immaturity as a prerequisite of viral entry into the brain parenchyma in congenital MCMV infection models, in which newborn mice are infected early postnatally.

While BBB properties start to establish early on, astrocyte differentiation spans the late embryonic and early postnatal phases (Ben-Zvi et al, 2014). Moreover, communication between brain endothelial cells and astrocytes seems to be vital to induce barrier functions (Rubin et al, 1991). Astrocytes secrete retinoic acid (RA), glial-derived neurotrophic factor (GDNF), sonic hedgehog (SHH) and angiopoietin 1 (Ang-1), which bind to endothelial receptors to increase junctional protein expression and in turn further decrease BBB permeability (Mizee et al, 2014; Igarashi et al, 1999; Alvarez et al, 2011; Pfaff et al, 2006; Suri et al, 1996). Thus, postnatal astroglial maturation along the BBB may prevent late CMV infections of the brain. CMV viremia may intervene with these processes and induce secondary signals (i.e., cytokines) that alter communication between astrocytes and endothelial cells (Fig. 4A), as shown for other viral infections. For example, immune-activated astrocytes secrete inflammatory cytokines, such as IL-6, which induces proteasomal degradation of the tight junction protein 1 in Japanese encephalitis virus cell culture experiments (Chang et al, 2015). TNF also promotes the production of matrix metalloproteinase (MMP) 9 in neurons and oligodendrocytes, which facilitates the disruption of the BBB (Kawasaki et al, 2015; Vafadari et al, 2016). Therefore, inflammatory signals from initial CMV entry into the fetus may disrupt the perinatal maturation of the BBB or even induce the breakdown of already established brain endothelial tight junctions (Fig. 4A). However, the effect of CMV infection on astrocytes needs to be experimentally addressed.

Scanning electron microscopy further revealed that the brain vessel lumen in neonatal mice appeared to be rougher and more porous following an MCMV infection (Kawasaki et al, 2017), suggesting that the damage to the vascular lumen potentially allows for access of MCMV to the CNS (Fig. 4B). The ability of MCMV to use endothelial cells as a site for initiating infection was supported by injection of MCMV into the superficial temporal vein in neonatal mice, where the majority of the MCMV protein IE1 was localized in CD31$^+$ endothelial cells and to a certain extent in pericytes (Kawasaki et al, 2015). Indeed, once endothelial barrier integrity is artificially disrupted via mannitol treatment, increased numbers of platelet-derived growth factor beta (PDGFRβ) and

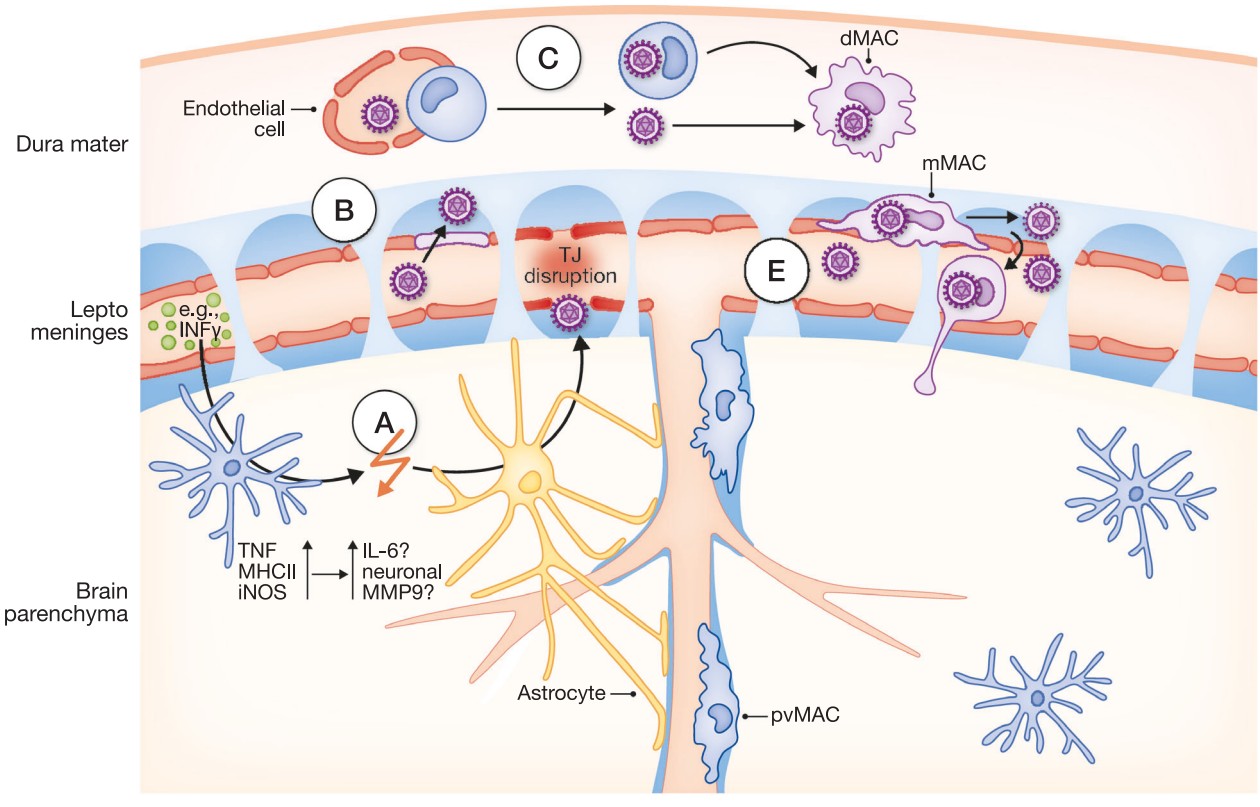

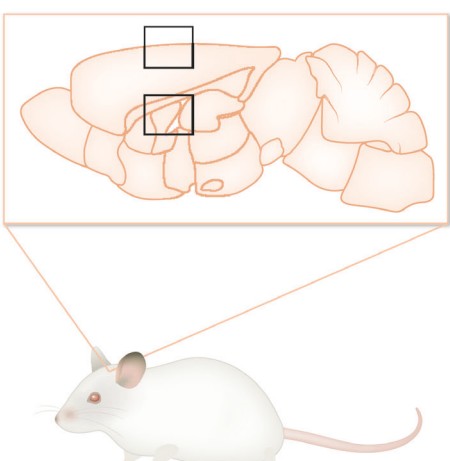

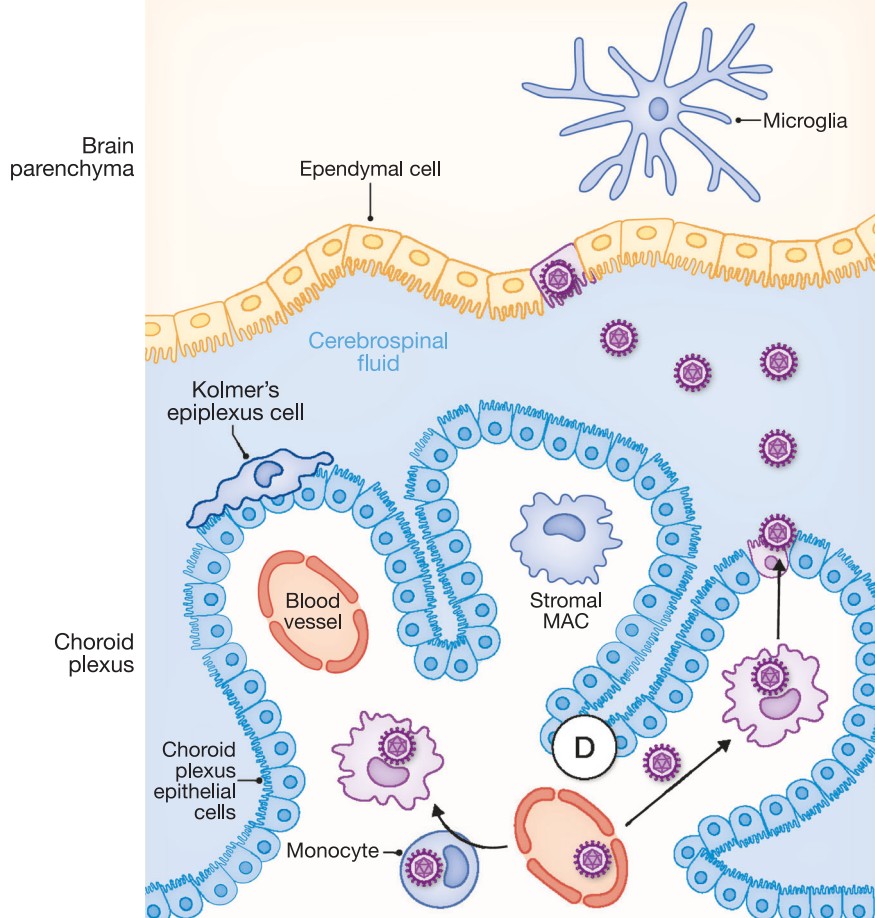

**Figure 4. Anatomical and cellular gateways for CMV to breach CNS barriers during fetal development.**

The brain parenchyma is surrounded by multiple layers of anatomical structures that separate the CNS from the periphery. These structures include the dura mater, the leptomeninges, and the choroid plexus within the ventricles. Once CMV reaches the fetal circulation, it will also infect monocytes in the circulation or nearby skull bone marrow. (A) CMV-induced IFN-γ activates microglia, leading to the secretion of TNF, which may disrupt astrocyte–endothelium communication or promote MMP9 production in neurons, thereby compromising tight junction integrity. (B) CMV infects endothelial cells directly. (C) Infected monocytes extravasate and differentiate into dMACs that support viral replication. Alternatively, cell-free virions infect dMACs directly. (D) Similarly, CMV reaches the choroid plexus through its fenestrated blood vessels, infecting stromal cpMACs and is ultimately shed to the CSF. (E) Meningeal MACs (and pvMACs) either reach into the vasculature where they can be directly infected via cell-free virus or become infected through viral particles in the CSF. Infected mMACs may be reprogrammed, disseminate CMV locally, and migrate towards the parenchyma. TJ tight junction.

neuron-glial antigen 2 (NG2) expressing pericytes were infected (Kawasaki et al, 2015).

Collectively, the brain is safeguarded by a multi-layered immune and structural barrier system, with the BBB as a key component in regulating CNS access. While functionally restrictive by late gestation, the BBB continues to mature postnatally, a window during which CMV may target endothelial cells and disrupt astrocyte-endothelial communication via the induction of inflammatory cytokines. CMV-induced endothelial damage may enable viral entry into the CNS perinatally, though it is unclear why cytokines or direct infection do not lead to BBB breakdown in adult mice.

### CNS-associated MACs

Among resident immune cells in the CNS interfaces, tissue-specific and adapted CNS-associated MACs (CAMs), which include subsets of choroid plexus MACs (cpMACs), leptomeningeal MACs (mMACs), dural MACs (dMACs), and perivascular MACs (pvMACs), are the most frequent (Kierdorf et al, 2019). They are positioned alongside the CNS interfaces, in close contact with blood vessels and in direct contact with the CSF (Kierdorf et al, 2019; Munro et al, 2022). Hence, CAMs are an inarguable part to provide immune protection and immune surveillance along the CNS barriers.

dMACs share a common progenitor with microglia (Kierdorf et al, 2013; Goldmann et al, 2016). In contrast to microglia, a subset of dMACs experiences an active turnover by monocyte influx throughout life. These monocytes migrate via direct ossified connections from the cranial bone marrow to the dura mater in a steady state (Cugurra et al, 2021; Herisson et al, 2018) giving rise to MHCII+ dMACs alongside embryonic MHCII- dMACs (Goldmann et al, 2016; Van Hove et al, 2019). Recently, dMACs have attracted increasing interest, assigning them a more important role in antiviral immunity than previously appreciated (Rua and McGavern, 2018; Van Hove et al, 2019). dMACs were shown to have acute antiviral functions, which are controlled by *Stat1* and *Ifnar1*, and prevent the invasion of vesicular stomatitis virus into the brain parenchyma (Mishra et al, 2020). Monocyte-derived MHCII+ dMACs that appear only in the postnatal period and whose frequency positively correlates with age (Rebejac et al, 2022), may link the susceptibility of newborns to MCMV encephalitis to the lack of MHCII+ dMAC. Indeed, in lymphocytic choriomeningitis virus infection, lower frequencies of MHCII+ dMACs are associated with a higher viral load (Rebejac et al, 2022). As introduced above, the location of the dura mater may allow privileged replenishment by monocytes directly from the skull (Cugurra et al, 2021). As MCMV has been shown to infect the bone marrow during acute

infection (Gibbons et al, 1994), the virus might piggyback these replenishment mechanisms by myeloid progenitors and further facilitate infection of dMACs. Moreover, under infectious conditions, dMACs receive monocyte input from the periphery in a CCR2-dependent manner (Gres et al, 2024). CMV-infected monocytes could also add to the dMAC pool, thereby causing meningitis. Lastly, the dura's fenestrated blood vessels could also enable free viral particles to infect dMACs directly (Fig. 4C).

Similar to the dura mater, the endothelium of the choroid plexus in the ventricular system of the brain is fenestrated and lacks tight junctions. Instead, choroid epithelial cells surround the blood vessels and stromal tissue of the choroid plexus and form tight junctions on their apical side (Alves De Lima et al, 2020). Stromal cpMACs are partially replenished by monocytes already in steady state (Prinz et al, 2021), while a small subset of intraventricular MACs (Kolmer's epiplexus cells) has self-renewal properties. Thus, MCMV could employ similar strategies to infect cpMACs via free virions that gain access through the fenestrated endothelium or via the replenishment of MACs by infected monocytes (Fig. 4D). Interestingly, HCMV has been found in the CSF of HIV-infected patients and congenitally infected infants (Long et al, 1998; Goycochea-Valdivia et al, 2017). Viral particles in the CSF may be derived from infected stromal cpMACs after subsequent infection of choroid epithelial cells (Fig. 4D).

pvMACs colonize the emerging perivascular space along the brain arteries originating from a pool of mMACs, as soon as this anatomical niche forms (postnatal day 5-14 in mice and after gestational week 25 in humans) (Masuda et al, 2022). Notably, most data originate from mice infected with MCMV prior to postnatal day 5, at which pvMACs along the brain vasculature are still absent. Accordingly, the initial cellular CMV target cell in the brain may depend on the time of infection, e.g., infections in the first or second pregnancy trimester may particularly affect neural stem cells (Liu et al, 2017), whereas later infections may spread after initial replication in highly differentiated pvMACs. pvMACs potentially mitigate the infection, analogous to the respiratory tract infection of adult mice (see the section "The lung"). Once present, their positioning renders pvMACs a potential primary target when CMV viremia occurs, since it has been proposed that pvMACs have direct access to the vascular lumen (Masuda et al, 2022). This concept is in agreement with the inverse correlation between neurological complications after infections and the gestational age (Moulden et al, 2021). In addition, pvMACs regulate BBB permeability and drainage of the brain parenchyma similarly to their counterparts in peripheral organs, where vessel-associated MACs determine vascular permeability (Coyne and Lazear, 2016; Van Hove et al, 2025). Thus, future studies would

need to dissect the differential impact of CMV infections in the CNS, carefully comparing infection time points at which pvMACs are absent or present.

Although under dispute, mMACs may also access the bloodstream with their protrusions, especially during inflammation (Barkauskas et al, 2013). Thus, mMACs could represent initial targets during CMV viremia and are putatively infected after intraplacental infection of fetal mice (Sakao-Suzuki et al, 2014). In this scenario, pvMACs and mMACs might be directly infected without the destruction of other cellular barrier components. These MACs could then directly promote local dissemination of CMV in the meninges and ultimately in the brain, as previously described in lungs (Baasch et al, 2021) (Fig. 4E). Moreover, mMACs that are located within the subarachnoid space are surrounded by CSF. As MCMV-infected MACs are long-lived and promote a full viral life cycle (Baasch et al, 2021), infected mMACs may release CMV particles into the CSF and propagate viral spread (Atkins et al, 1994) (Fig. 4E) to other regions of the brain. There, ependymal cells, which line the CSF-containing ventricular system of the brain, can be infected, as shown for HCMV or MCMV after intracranial mouse infection (Gabrielli et al, 2009; Van Den Pol et al, 2002). Thus, CMV may also enter the brain at multiple locations simultaneously via either viral particles or infected cells in the CSF.

Lastly, it is important to mention that the nasal cavity contains a BBB analog, the blood-olfactory barrier, which separates blood vessels from the olfactory sensory neurons that detect odors. Antibodies from the bloodstream can enter the respiratory mucosa in the nose, while they cannot enter the olfactory mucosa which connects the respiratory tract and the CNS by the olfactory sensory neurons (Wellford et al, 2022). Thus, the olfactory barrier functions as an immune barrier, but its structure is not fully understood. The olfactory system is also targeted frequently in congenital HCMV infections (Teissier et al, 2014), and olfactory neurons and MACs of the nasal epithelium have also been shown to be infected with MCMV in neonatal mice (Farrell et al, 2016b).

### Brain parenchyma and microglia

Microglia are the resident MACs of the brain parenchyma. They originate during early embryogenesis (Kierdorf et al, 2013), where they clear apoptotic or excess cells and remodel neural circuits through synaptic pruning (Prinz et al, 2019). As central immune effector cells of the CNS parenchyma, microglia respond to alterations in the CNS environment by modifying their morphology, proliferative capacity, and transcriptomic as well as secretory profile (Prinz et al, 2019). Accordingly, during neonatal MCMV infection, microglia are activated (Kveštak et al, 2021; Kosmac et al, 2013; Schwabenland et al, 2023), characterized by TNF production, expression of MHCII and inducible nitric oxide synthase (iNOS), and local proliferation (Kveštak et al, 2021) (Fig. 4A). As mentioned above, TNF secretion by microglia could further disrupt BBB integrity and enable direct viral invasion or immune cell infiltration. Accordingly, IFNγ produced by peripheral CD4⁺ and CD8⁺ T cells in the spleen can activate microglia, leading to the loss of excitatory synapses (Schwabenland et al, 2023). In addition, during later stages of neonatal MCMV infection (8–11 days post-infection), IFNγ from brain-infiltrating NKp46⁺ innate lymphoid cells/NK cells drives microglial activation and has been shown to alter cerebellar development (Kveštak et al, 2021). Despite the detrimental effects in the developing brain, IFNγ secretion along with IL-12-signaling is

required for the resolution of acute MCMV infection in neonatally infected mice, as evidenced by sustained viral replication in the brain up to 70 days post infection in $Ifng^{-/-}$ and $Il12rb2^{-/-}$ mice (Krstanović et al, 2025). Yet, treatment with anti-IFNγ antibodies throughout the infection reduces microglia activation and corrects the increased size of the external granular layer, the initial position of granule neuron progenitor cells at birth, in the cerebellum after MCMV infection (Kveštak et al, 2021). Comparison of neuron- (BAF53b^cre^), astrocyte- (*Gfap*^cre^) or myeloid cell (*Lyz2*^cre^)-specific knockout of IFNγ signaling (*Ifngr1*^floxed^) after MCMV infection revealed amelioration of a dysregulated Shh pathway only in mice that lack the *Infgr1* in neurons (Kveštak et al, 2021), indicating complex (microglia- and neuron-mediated) effects of IFNγ-mediated neuronal damage. It is worth noting, however, that *Lyz2* targets less than 50% of spinal cord microglia (Goldmann et al, 2013), which may limit its utility as a model for studying microglia in CNS infection experiments. Chemical depletion of microglia by intracerebroventricular injection of clodronate in fetal rats resulted in higher survival, less neuropathology, and better cognitive function after birth (Cloarec et al, 2018), pointing to a pivotal role of microglia in CMV pathogenicity.

Neurons in the brain also represent a reservoir for latent MCMV infection, and reactivation is controlled by CD4⁺ (Krstanović et al, 2025) and CD8⁺ T cells (Brizić et al, 2018). Interestingly, the hippocampus appears to be a major brain area for reactivation in neonatal MCMV (Krstanović et al, 2025), as well as congenital HCMV (Piccirilli et al, 2023) infections.

In summary, the BBB is likely mature enough to prevent direct viral invasion during the third trimester in humans and in early postnatal infection models in mice. However, CMV can directly infect brain endothelial cells, disrupting tight junction integrity and compromising the BBB. Furthermore, CMV infection activates microglia that contribute to neurodevelopmental defects, but may also facilitate further breakdown of the BBB during the course of infection, as indicated via lymphocyte invasion into the brain. It seems conceivable that activated microglia can act as antigen-presenting cells after CMV brain infection and propagate a vicious circle by inducing IFNγ producing T cells, although firm evidence is currently missing.

CAMs, located at brain interfaces and near blood vessels, remain largely understudied in CMV infection models but may play a pivotal role during fetal and early postnatal infection. The dura mater and choroid plexus, both containing fenestrated vasculature, are particularly noteworthy, as these regions may serve as key entry points for viral particles and infected cells into the CNS.

Yet, under the assumption that CMV has the capacity to infect endothelial cells and activate microglia, CMV does not reach the brain in immunocompetent adult humans or mice. It is well established that embryonic and neonatal microglia are fundamentally distinct from their adult counterparts, characterized by an amoeboid morphology, high proliferative activity, and roles in tissue remodeling (Prinz et al, 2019). Another distinction between neonatal and adult CNS immune cell composition is the timing of pvMAC seeding, which occurs during late embryogenesis (human) or after birth (mice) through a leptomeningeal MAC progenitor (Masuda et al, 2022). Given the complexity and diversity of myeloid populations in the CNS during development, a detailed profiling of CNS immune populations in infection and across developmental stages is essential to improve our knowledge of CMV pathogenesis.

## CMV-Encephalitis in adult patients under immunodeficient/-suppressive conditions

Postnatally acquired CMV infection does not affect the brain in immunocompetent individuals. On the other hand, immunocompromised or immunosuppressed patients are at high risk of developing CMV encephalitis upon infection. In HIV-infected individuals, for example, CMV is one of the most common opportunistic pathogens (Arribas et al, 1996). Historically, this affected up to 40% of patients with acquired immune deficiency syndrome (AIDS) (Díaz-Brochero et al, 2023). In particular, patients with very low CD4+ T-cell counts and signs of systemic HCMV infection may develop HCMV encephalitis (Mamidi et al, 2002). Similarly, the absence of an adaptive immune system in severe combined immunodeficiency (SCID) mice facilitates MCMV-induced encephalitis, while immunocompetent adult wild-type mice are resistant to it (Reuter et al, 2004). As HIV infects CD4+ helper T cells, one may speculate that it reduces the pool of CMV-specific T cells, leading to impaired control of CMV reactivation and higher viral load. In this scenario, the capacity of HCMV to infect the brain might be a sheer numbers game, in which the viral load potentially overwhelms the barriers of the brain. In contrast to this hypothesis, increased numbers of HCMV-specific memory T cells have been observed in the blood of HIV and HCMV double seropositive patients (Abana et al, 2017; Komanduri et al, 2001). Upon in vitro reactivation, isolated memory T cells from these patients secrete increased levels of inflammatory cytokines, which may lead to endothelial damage (Abana et al, 2017) and thus BBB breach. Accordingly, about 50% of AIDS patients and up to 100% of HIV-associated dementia patients exhibit increased BBB permeability (Calcagno et al, 2014). Although characteristic features of HCMV encephalitis are particularly hard to decipher in co-infections, HIV infections serve as an example of how (acquired or inborn) immunodeficiency in mice and humans can help to dissect factors that contribute to the infection of protected organs, such as the brain.

## Conclusion

CMV infections are complex due to the virus's broad host cell tropism, lifelong persistence, and ability to alternate between active, latent, and reactivated infection states. A key factor in establishing productive infection at mucosal surfaces is the virus's interaction with myeloid immune cells, especially DCs and MACs, which are particularly abundant in barrier tissues. CMV appears to exploit (i) resident MACs supporting viral replication, and reprogramming them for local dissemination (Baasch et al, 2021), (ii) DCs to reach the SG as a site of infection and (iii) circulating monocytes by inducing their recruitment towards site of infection, promoting either their differentiation (Smith et al, 2004a) or further viral dissemination. This strategy may reflect a co-evolutionary adaptation at mucosal interfaces, allowing CMV to penetrate barriers, such as the lung without triggering overt pathology in immunocompetent hosts (Box 1). Furthermore, CMV's activation of proximal bystander at the site of infection and distal progenitor cells in the bone marrow positions it to modulate immune responses from early life onward. It is feasible that CMV might

have developed a common mechanism to manipulate and exploit myeloid cells in order to surpass distinct barrier tissues in the entire body. However, there is also evidence for distinct strategies in the placenta and in the fetal/neonatal brain (Box 1). Age-dependent implications for CMV infection may be influenced by the presence or maturity of structural cells and MACs in barrier organs, including the lung and potentially the brain. Despite significant progress in our understanding of CMV immunity in adults, a substantial gap remains in our knowledge of how CMV shapes and is shaped by the unique immune environment of the fetus and neonate (Box 1). While advances in multidimensional immune profiling offer a promising path forward, a more focused effort to decipher perinatal immunity is urgently needed to define the early host-virus interplay and to inform targeted strategies aimed at preventing CMV-associated pathology during this vulnerable developmental window.

## Peer review information

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

## Acknowledgements

PK was funded by DFG KO6815/1-11. RI received funding by the DFG TRR359 (Project ID 491676693, LAUNCH1). ZR received funding by the DFG TRR359 (Project ID 491676693). KK was supported by the DFG by project grants within the TRR359 (Project ID 491676693), TRR167 (Project ID 259373024), CRC1479 (Project ID 441891347), CRC1160 (Project ID 256073931), RU by the DFG under Germany's Excellence Strategy (grant no. CIBSS—EXC-2189, Project ID 390939984) and by the Heisenberg program of the DFG (Project ID 544402801). PH received funding by the DFG TRR359 (Project ID 491676693), TRR167 (Project ID 259373024) and CRC1160 (Project ID 256073931). SB was funded by the DFG TRR359 (LAUNCH1) and the Hans A. Krebs Medical Scientist Program (Faculty of Medicine, University of Freiburg).

## Author contributions

**Agnibesh Dey**: Data curation; Validation; Writing—original draft; Writing—review and editing. **Vitka Gres**: Data curation; Validation; Visualization; Writing—original draft; Writing—review and editing. **Alina Nelipovich**: Validation; Writing—original draft; Writing—review and editing. **Philipp Kolb**: Validation; Writing—original draft; Writing—review and editing. **Roland Immler**: Validation; Writing—original draft; Writing—review and editing. **Zsolt Ruzsics**: Validation; Writing—original draft; Writing—review and editing. **Katrin Kierdorf**: Validation; Writing—original draft; Writing—review and editing. **Philipp Henneke**: Conceptualization; Supervision; Validation; Writing—original draft; Writing—review and editing. **Sebastian Baasch**: Conceptualization; Supervision; Validation; Visualization; Writing—original draft; Writing—review and editing.

## Disclosure and competing interests statement

The authors declare no competing interests.

