## [Peer Review File · EMBO Reports]

Breaking barriers: How cytomegalovirus manipulates myeloid cells to invade tissues

Agnibesh Dey, Vitka Gres, Alina Nelipovich, Philipp Kolb, Roland Immler, Zsolt Ruzsics, Katrin Kierdorf, Philipp Henneke, and Sebastian Baasch

Corresponding author(s): Sebastian Baasch (sebastian.baasch@uniklinik-freiburg.de)

Review Timeline:

Submission Date:	22nd Sep 25
Editorial Decision:	20th Oct 25
Revision Received:	8th Feb 26
Accepted:	24th Mar 26

Editor: Achim Breiling

Transaction Report:

Dear Dr. Baasch,

Thank you for the submission of your review article to our editorial offices. I have now received the full set of referee reports that is copied below. As you will see, all three referees state that your manuscript is interesting and timely. However, they have several suggestions to improve the submission that I kindly ask you to address in a revised manuscript.

Given the constructive referee comments, I would thus like to invite you to revise your manuscript with the understanding that all referee points will be addressed in the revised manuscript and in a detailed point-by-point response.

I further have these editorial requests:

- Please provide a comprehensive final title with not more than 100 characters (including spaces).
- Please reduce the number of keywords to five.
- We have space for 1 more figure, and it would be important to have indeed 4 figures, as we encourage authors to maximize the use of visual elements, which will increase the accessibility of the piece to a non-specialist readership. Please consider adding 1 more figure and note the instructions regarding figures below.
- We usually ask our authors to include a box called "In need of answers" that briefly outlines the major questions that are still open in a given field in the form of a few bullet points. These questions can be accompanied by a brief explanation of what would be needed to address them and may provide helpful towards setting the stage for future experimentation in the field. For an example see this recent review we published: <https://www.embopress.org/doi/full/10.1038/s44319-024-00135-4>
- Please also add callouts for the box to the manuscript text (Box 1).
- We now use CRediT to specify the contributions of each author in the journal submission system. CRediT replaces the author contribution section. Please use the free text box to provide more detailed descriptions and do NOT provide your final manuscript text file with an author contributions section. See also our guide to authors: <https://www.embopress.org/page/journal/14693178/authorguide#authorshippinguidelines>
- Please use for the references and their callouts our reference format (with et al. for manuscripts with more than 10 authors): <http://www.embopress.org/page/journal/14693178/authorguide#referencesformat>
- Please make sure that all the funding information is also entered into the online submission system and that it is complete and similar to the one in the acknowledgement section of the manuscript text file.

I think this is a very interesting review and while I appreciate that incorporating the referees' suggestions will still require some work, I am convinced that the article is worth it and will benefit from it.

When submitting your revised manuscript, we will require a Microsoft Word file (.doc) of the revised manuscript text including detailed figure legends (at the very end), but without the figures.

Please provide the final figures as separate, high-resolution files as .pdf, .eps, .tif, or .jpg (one file per figure). Please finalize the drafts provided and make sure they accurately illustrate the key scientific concepts that you wish to show.

Please also note the following points:

- If there are certain aspects of your figure draft that are based upon assumptions or where the scientific data remains ambiguous (for example, schematically depicting a presumed direct protein-protein interaction, protein shape or subcellular localizations etc.) please add a comment so that we can work with you on an accurate depiction. Please ensure the directionality and nature of interactions is presented accurately.
- If the figure or single panels of the figure have been adapted from a published figure, please add this information to the figure legend (e.g., 'Adapted from...' or 'Based on...'). The editor will discuss if a reference and permission will be necessary
- Please only re-use figures or parts of a figure if this is essential for understanding the concept communicated. Often a reference to a previous paper will suffice. If the figure contains re-used images or elements of images, including schematics, micrographs or photos, please make sure that you have the permission/license to publish it (this also applies to your own

previous work, if the journal you published in retains copyright. Certain 'creative commons' open access licenses, such as CC-BY 4.0, allow re-use without additional formal permissions). All re-used material must be explicitly cited.

- If you use an image data base for scientific iconography (e.g., BioRender), please let us know if you have a license that allows for publication in an academic journal. Often authors use misleading iconography for expedience. Please ensure the information shown is scientifically accurate. If in doubt, please discuss with the editor or provide a sketch so that our designers can create accurate iconography.

- For figures created using a software for editing vector objects like Inkscape, CorelDraw etc., please send the file as a PDF (or SVG, or EPS), PowerPoint or Keynote in which the labels and objects are still editable. For figures created using Adobe Illustrator, please send the Illustrator (.ai) file.

I look forward to seeing a revised version of your manuscript when it is ready. Please let me know if you have questions or comments regarding the revision.

Kind regards,

Achim

Referee #1:

In this review, Dey and colleagues review the literature on cytomegalovirus (CMV) manipulation of myeloid cells during postnatal infection. They are particularly focused on the virus's ability to breach barrier tissues, including the placenta and brain.

While the topic is timely, the review reads very list-like and often lacks a take-home message or hypotheses from the authors based upon coalescence of published findings.

There are also places in the review, including in the introduction to myeloid cell infection, that should be edited for accuracy given the vast published data in this area.

While there is a focus on murine data, it would also be beneficial to complement such sections with additional data in other settings (e.g., guinea pig, human, organoid, etc).

Additionally, the manuscript requires a thorough review for language and grammar.

Overall, while a timely topic, the manuscript requires significant refinement and needs to be edited for accuracy and clarity.

Referee #2:

This is a very nice, interesting, and timely review which is structured very clearly and written very well. I have a few comments that I am listing below that should be addressed in a revision.

Part 2 - Myeloid lineage + CMV:

1st paragraph:

- Citation 11: Schulz et al 2012 does not fit here.
- "After birth, some tissues experience a turnover of embryonic MAC by monocyte-derived MAC, i.e., cells originating from hematopoietic stem cells (HSC) in the bone marrow [6]." -> citation 6 only refers to skin. Can the authors please provide (an) additional citation(s)?

2nd paragraph:

- Citation 20 does not fit here.

Part 3 - Breaching Barriers for postnatal infection

3.1 (the lung):

- Please provide citations in the first paragraph (especially regarding structure of respiratory tract and types of AECs).

Figure 1:

- In the text the authors refer to the (v)m131/129 protein as Mck2 and argue that the complex of Mck2, gH and hL is necessary to infect MACs, while gO is necessary to infect AECs. Moreover, preferential infection of MACs limits viral replication but promotes viral spread and vice versa. This relationship should be shown in the figure as well. It would also be nice to clarify that the proteins mentioned adjacent to the arrows are the mediating factors and that, for example, without gH, gL or Mck2, there is no MAC infection.
- The authors provide a detailed explanation in the written text, however the figure is not self-explanatory. It's unclear where the infected DCs come from and how they are different from MACs in the dissemination process. There is only one arrow leading from both infected MACs and DCs to the salivary gland (SG) or the lymph node and there are three factors mentioned. It should be clear that the DCs are migrating to the SG, which is mediated by CD44 and vM33, while MACs migrated to the draining lymph node in a Zeb1 controlled manner.
- Personal opinion: the lymph node looks like a bit like a speech bubble.
- Legend for cell types: the abbreviation "AEC" is not really needed next to the cell symbol.
- The symbol for the human being is a man (although I noted the slim waist) - can the authors pick a symbol that is even more gender unspecific? I would highly appreciate that :0).
- Figure legend: "In the lung, CMV targets cells such as..." ("s" is missing).

3.2 (the intestine):

1st paragraph:

- Missing citation for first paragraph.
- "...while Paneth cells in the small intestine secrete antimicrobial peptides." ("e" is missing).

Part 4 - Breaching barriers for prenatal infection

4.1 (the placenta):

Figure 2:

- To make the figure more self-explanatory, can the abbreviations FP, Ch, IVS, Dec, BV, SA be written out? Space for it would be there.
- Why are there two cell symbols for maternal MAC?

2nd paragraph (page 12):

- Citation 73 does not fit here, needs a different citation.

4th paragraph (page 13):

- introduction of hyperimmunoglobulin (HIG) treatment as abbreviation.

5th paragraph (page 14):

- Wrong reference to figure 2, it should be figure 2c instead of 2d: "Antibody-mediated transcytosis of free CMV through STB, damage of the STB layer, followed by infection of Hofbauer cells (Figure 2a and b) or induced migration of infected maternal MACs towards the fetal placenta may be employed (Figure 2d)."

4.2 (the CNS):

CNS barriers:

1st paragraph (page 16):

- Citation 100 does not fit here, a different citation is needed

2nd paragraph (page 16):

- The first citation (106) is not sufficient for all the previous information; another citation is needed (prob. 105).

Figure 3:

- A: the connection of the CMV infection, shown directly adjacent to the letter a, to the resulting inflammatory signals that activate astrocytes that in turn lead to proteasomal degradation of the tight junctions is not clear. The four parts of the figure (CMV infection; microglia and astrocyte activation by inflammatory cytokines resulting from the CMV infection; production of additional cytokines + activation of MMP9; degradation of tight junctions) should be connected better to show the step-by-step progression. Additionally, the tight junction disruption symbol should be changed to a different color, as it is nearly not visible on the red background.
- E: the part e is mentioned in the figure legend but is not shown in the figure itself or in the text.
- BV (blood vessel), CSF (cerebrospinal fluid) and Ven (ventricular system) as abbreviations are missing; the abbreviations for Ep (ependymal cell layer) as well as CP (choroid plexus) are mentioned in the legend but do not occur in the figure. I would suggest to write out most of these abbreviations in the figure to make the figure more self-explanatory.

CNS-associated MACs:

2nd paragraph (page 20):

- "The most abundant immune cells in the dura are dural MACs that experience an influx of monocytes" - Which influx? Do the dural MACs result from this influx or is the influx of those monocytes additional to the already existing dMACs?

3rd paragraph (page 21):

- Missing s: "Stromal cpMACs are partially replenished by monocytes already in steady state [128]"

4th paragraph (page 21):

- "Although under dispute, mMACs may also access the bloodstream with their processes, especially during inflammation [135]"
- What do the authors mean when writing about "processes"?

Referee #3:

This review article, entitled „Breaking barriers - How cytomegalovirus manipulates myeloid cells to establish an infection and enter secluded tissues", by Agnibesh Dey and colleagues provides a very informative and timely overview of CMV infection of myeloid cells and how this helps the virus to overcome tissue barriers.

The manuscript is very well written and easy to understand. It fills a gap as there is no other recent review with a similar focus. Below are a few suggestions for the authors' consideration.

Major issues:

1. MCMV infection of mice serves as a convenient and tractable model to study CMV pathogenesis in vivo. There are many similarities between mice and humans, and MCMV and HCMV. However, there are also differences, which should not be swept under the rug. In most parts of this review, the authors treat the two systems as equal and mix findings obtained in the mouse system with those obtained in human patients. In my opinion, this is not appropriate. Therefore, I suggest that the authors separate the findings from the two systems in each chapter. In many chapters, the results rely primarily on the MCMV/mouse system. This could be stated at the beginning of the chapter. Towards the end of the chapter, the authors could summarize the findings from the human system and discuss how they agree (or disagree) with the findings from the mouse system. In a few cases, the current state of knowledge is largely based on data from human studies (e.g., GI infections, encephalitis in adults). Here, the authors could turn it around and first report the findings from the human system before comparing them to those from the mouse system.

2. Myeloid cells, particularly DCs and MACs, play an important role in the adaptive immune response. Surprisingly, this aspect is barely covered in this review. It would be fair to devote at least one paragraph to the interaction between myeloid and lymphoid cells and how this interaction is modulated by HCMV and MCMV. The authors could direct the reader to other review articles if they feel that detailed information would go beyond the scope of the present review.

Minor points:

3. The authors should mention in the abstract that results from both human and murine systems are being reviewed and discussed.

4. In some instances, general statements are not underpinned by appropriate citations. In the Introduction, for instance, reference #1 is not ideal to support the statements. More recent and more focused reviews are available. The same is true for reference #2. There are no references at all for the statements in the remainder of the paragraph.

5. Introduction, page 2. "microbial rich and immunologically demanding" -> please rephrase.

6. Page 3. "manipulating structural and especially myeloid immune cells". Please define structural immune cells.

7. Page 4. It is not true that M36 "specifically prevents programmed cell death in MAC", although it is crucial for viral replication in MAC in vitro and dissemination in vivo. However, there are other MCMV and HCMV cell death suppressors of similar importance for MAC infection, which were not mentioned.

8. Page 8. "Our previous study identified". One should generally not write review articles using the first person, as academic and formal writing emphasizes objectivity by avoiding "we" and "our". Instead, one should focus on the literature and findings, use the third-person or passive voice, and state conclusions directly without centering them on one's personal perspective.

9. Figure 1. It is not helpful to coin new abbreviations that are not used in the literature. The commonly used abbreviation for the viral glycoprotein O is gO, not vgO, and the M33 protein is usually called M33 or pM33, but not vM33.

10. Page 11. "Intragastral infection" -> gastric or intragastric (?)

11. Page 13. The "evidence supporting IgG-dependent transcytosis via FcRn across the STB layer" needs a better explanation. If FcRn facilitate transcytosis of viruses bound antibodies, why aren't all viruses with a viremic phase transmitted to the fetus? Does this imply that non-neutralizing HCMV-specific antibodies facilitate transmission across the STB layer?

Achim Breiling, PhD
Senior Scientific Editor, EMBO reports

Institute for Infection Prevention
and Control

Sebastian Baasch, DVM

sebastian.baasch@uniklinik-freiburg.de

Freiburg, February 6th 2026

Resubmission of “Breaking barriers: How cytomegalovirus manipulates myeloid cells to invade tissues” (EMBO-2025-62725V2) to *EMBO reports*

Dear Achim Breiling,

We resubmit the manuscript “Breaking barriers: How cytomegalovirus manipulates myeloid cells to invade tissues” to *EMBO reports* after substantial revision and the inclusion of several additional data and a figure.

We appreciate that the referees found our review manuscript timely (all 3 referees), interesting (referee #2) and unique (referee #3) with a good structure (referee #2-3) and easy to read and understand (referee #2-3). Nonetheless, we agree with the referees and the points raised are valid and warrant careful consideration. Below, we address each of the referees' comments point by point.

Key changes in the new version of the manuscript include:

1. Inclusion of multiple CMV species additional to mouse CMV and human CMV, such as guinea pig CMV and non-human primate CMV. Moreover, we added data obtained from organoids in the respective experimental system.
2. We have substantially improved the text, correcting spelling or grammatical errors and revising overly long sentences to improve clarity.
3. We have added additional references to acknowledge previous work and back up statements.
4. We have added a Figure for the section “The intestine”.

We are convinced that these changes improve both the validity and readability of this review manuscript. We address each reviewer comment in the attached point-by-point response.

We are grateful to the reviewers for their valuable comments and hope that the revised manuscript will meet with the approval of *EMBO reports*.

Sincerely,

Sebastian Baasch

Point-by-point response

Referee #1:

While the topic is timely, the review reads very list-like and often lacks a take-home message or hypotheses from the authors based upon coalescence of published findings.

We appreciate that the reviewer acknowledges our review topic as timely. At the same time, we regret that the reviewer feels that our review reads very list-like. However, we explore many relevant tissues in the context of initiation of a cytomegalovirus infection. This requires that every tissue and its respective barrier needs a short introduction (i.e. anatomy, relevant cell types involved, etc.), followed by elucidation of known or putative mechanisms of viral invasion. The repetitive style might appear redundant and “list-like”, but states a necessity to understand whether and how CMV breaches the barrier tissue. We also hope that this enables the reader that is only interested in a certain organ to fast forward to their tissue of interest.

Similarly, we provide at the end of each section a summary of the previous detailed elaboration, additionally weigh up the likelihood of the respective strategy and highlight needs for future research.

Altogether, we hope that given these explanations the reviewer understands and appreciates the structure and the content of our manuscript.

There are also places in the review, including in the introduction to myeloid cell infection, that should be edited for accuracy given the vast published data in this area.

We acknowledge the amount of previous work that has been done in the context of barrier tissues and infection with CMVs and tried to reference accordingly. In the “myeloid cell infection” section, we have now edited the title to state the focus on replication, dissemination and latency.

Moreover, for clarity and to avoid redundancy, studies such as those on lung CMV infection are not discussed in this section but are instead covered in the “Lung” section. Due to the scope of this review, we do not discuss, for example, neutrophils or the interaction with lymphocytes.

Nonetheless, if the reviewer thinks we have still missed, a crucial reference, we are grateful if the reviewer could name it.

While there is a focus on murine data, it would also be beneficial to complement such sections with additional data in other settings (e.g., guinea pig, human, organoid, etc).

We have now added suitable information about mouse, human, rat, guinea pig, non-human primates and organoids if available and in the scope of the review.

Additionally, the manuscript requires a thorough review for language and grammar.

We have critically read through the manuscript and corrected spelling or grammatical errors and revised overly long sentences to improve clarity.

Referee #2:

Part 2 - Myeloid lineage + CMV:

1st paragraph:

- Citation 11: Schulz et al 2012 does not fit here.

We appreciate the thorough revision and apologize for the flaw.

We have now referenced the correct work by

Schulz, C., Perdiguero, E. G., Chorro, L., Szabo-Rogers, H., Cagnard, N., Kierdorf, K., ... & Geissmann, F. (2012). A lineage of myeloid cells independent of Myb and hematopoietic stem cells. *Science*, 336(6077), 86-90.

- "After birth, some tissues experience a turnover of embryonic MAC by monocyte-derived MAC, i.e., cells originating from hematopoietic stem cells (HSC) in the bone marrow [6]." -> citation 6 only refers to skin. Can the authors please provide (an) additional citation(s)?

The reviewer is correct. We state that some tissue experience the turnover by monocyte and, thus, should reference >1 publications. We have now added examples from the heart, intestine and dura:

Molawi, K., Wolf, Y., Kandalla, P. K., Favret, J., Hagemeyer, N., Frenzel, K., ... & Sieweke, M. H. (2014). Progressive replacement of embryo-derived cardiac macrophages with age. *Journal of Experimental Medicine*, 211(11), 2151-2158.

Bain, C. C., Bravo-Blas, A., Scott, C. L., Gomez Perdiguero, E., Geissmann, F., Henri, S., ... & Mowat, A. M. (2014). Constant replenishment from circulating monocytes maintains the macrophage pool in the intestine of adult mice. *Nature immunology*, 15(10), 929-937.

Cugurra, A., Mamuladze, T., Rustenhoven, J., Dykstra, T., Beroshvili, G., Greenberg, Z. J., ... & Kipnis, J. (2021). Skull and vertebral bone marrow are myeloid cell reservoirs for the meninges and CNS parenchyma. *Science*, 373(6553), eabf7844.

2nd paragraph:

- Citation 20 does not fit here.

We appreciate the thorough revision and apologize for the mistake.

We have now referenced the correct work by

Ebermann, L., Ruzsics, Z., Guzmán, C. A., van Rooijen, N., Casalegno-Garduño, R.,

Koszinowski, U., & Čičin-Šain, L. (2012). Block of death-receptor apoptosis protects mouse cytomegalovirus from macrophages and is a determinant of virulence in immunodeficient hosts. *PLoS Pathogens*, 8(12), e1003062

Part 3 - Breaching Barriers for postnatal infection

3.1 (the lung):

- Please provide citations in the first paragraph (especially regarding structure of respiratory tract and types of AECs).

We thank the reviewer for pointing out the lack of references. Specifically, we have now corrected the claim that ~90% of the AECs are AEC type I. Indeed, the cited proportion is the surface of the alveolus that the AEC I cover, while the ratio of AECI to AECII is roughly 1:1. We have added appropriate references.

Stone, K. C., Mercer, R. R., Gehr, P., Stockstill, B., & Crapo, J. D. (1992). Allometric relationships of cell numbers and size in the mammalian lung. *Am J Respir Cell Mol Biol*, 6(2), 235-43.

Guillot, L., Nathan, N., Tabary, O., Thouvenin, G., Le Rouzic, P., Corvol, H., ... & Clement, A. (2013). Alveolar epithelial cells: master regulators of lung homeostasis. *The international journal of biochemistry & cell biology*, 45(11), 2568-2573.

Westphalen, K., Gusarova, G. A., Islam, M. N., Subramanian, M., Cohen, T. S., Prince, A. S., & Bhattacharya, J. (2014). Sessile alveolar macrophages communicate with alveolar epithelium to modulate immunity. *Nature*, 506(7489), 503-506.

Hussell, T., & Bell, T. J. (2014). Alveolar macrophages: plasticity in a tissue-specific context. *Nature reviews immunology*, 14(2), 81-93.

Figure 1:

- In the text the authors refer to the (v)m131/129 protein as Mck2 and argue that the complex of Mck2, gH and hL is necessary to infect MACs, while gO is necessary to infect AECs. Moreover, preferential infection of MACs limits viral replication but promotes viral spread and vice versa. This relationship should be shown in the figure as well. It would also be nice to clarify that the proteins mentioned adjacent to the arrows are the mediating factors and that, for example, without gH, gL or Mck2, there is no MAC infection.

We are grateful for the reviewer's thoughtful ideas. We have now highlighted the role of the different entry complexes and also specified that M33 and CD44 is required for the DC-mediated dissemination from the lymph node to the salivary gland and not from the lung to the lymph node. Although we strongly believe that immunity against CMV differs substantially between

neonates/infants and adults, we have now more cautiously highlighted in the text the differences in the depletion models used in studies of neonatal and adult mice. Accordingly, we decided not to include this comparison in the figure.

- The authors provide a detailed explanation in the written text, however the figure is not self-explanatory. It's unclear where the infected DCs come from and how they are different from MACs in the dissemination process. There is only one arrow leading from both infected MACs and DCs to the salivary gland (SG) or the lymph node and there are three factors mentioned. It should be clear that the DCs are migrating to the SG, which is mediated by CD44 and vM33, while MACs migrated to the draining lymph node in a *Zeb1* controlled manner.

Thank you for highlighting this. We have described in the figure legend and the main text that DCs reside in close proximity to the alveolar epithelium, and we hope the reviewer considers this explanation sufficient. We have now improved visualization and clarified that infected AMs migrate to the lymph node in a *Zeb1*-dependent fashion, while DCs enter the lymph node but only proceed to the salivary gland in the presence of M33 and CD44.

- Personal opinion: the lymph node looks like a bit like a speech bubble.

The reviewer is right, the schematic needed more details to appear reminiscent of a lymph node. We added now afferent lymph vessels.

- Legend for cell types: the abbreviation "AEC" is not really needed next to the cell symbol.

We have now removed the abbreviation "AEC".

- The symbol for the human being is a man (although I noted the slim waist) - can the authors pick a symbol that is even more gender unspecific? I would highly appreciate that :0).

The reviewer is right. The symbol appeared to have rather male features. We have now narrowed the shoulder breadth and chin line.

- Figure legend: "In the lung, CMV targets cells such as..." ("s" is missing).

We have added the "s".

3.2 (the intestine):

1st paragraph:

- Missing citation for first paragraph.

We have now added the appropriate reference

T. Ayabe *et al.* Secretion of microbicidal alpha-defensins by intestinal Paneth cells in response to bacteria. *Nature Immunology* **1**, 113–118 (2000).

- "...while Paneth cells in the small intestine secrete antimicrobial peptides." ("e" is missing).

We have added the "e".

Part 4 - Breaching barriers for prenatal infection

4.1 (the placenta):

Figure 2:

- To make the figure more self-explanatory, can the abbreviations FP, Ch, IVS, Dec, BV, SA be written out? Space for it would be there.

We added and refined the full description of the structures.

- Why are there two cell symbols for maternal MAC?

The reviewer is correct, two symbols for maternal MAC are confusing. In the previous figure, we have depicted the maternal decidual MAC alongside the placenta-associated maternal MAC for the sake of completeness. However, we have not discussed the decidual MAC in the manuscript. We have now removed the decidual MAC and its symbol from the figure.

2nd paragraph (page 12):

- Citation 73 does not fit here, needs a different citation.

We appreciate the thorough revision and apologize for the mistake.

We have now referenced the correct work by

Aplin, J. D. (2010). Developmental cell biology of human villous trophoblast: Current research problems. *The International Journal of Developmental Biology*, 54(2–3), 323–329.

<https://doi.org/10.1387/ijdb.082759ja>

Robbins, J. R., Skrzypczynska, K. M., Zeldovich, V. B., Kapidzic, M., & Bakardjiev, A. I. (2010).

Placental syncytiotrophoblast constitutes a major barrier to vertical transmission of *Listeria monocytogenes*. *PLoS Pathogens*, 6(1), e1000732. <https://doi.org/10.1371/journal.ppat.1000732>

4th paragraph (page 13):

- introduction of hyperimmunoglobulin (HIG) treatment as abbreviation.

We have now introduced the abbreviation properly.

5th paragraph (page 14):

- Wrong reference to figure 2, it should be figure 2c instead of 2d: "Antibody-mediated transcytosis of free CMV through STB, damage of the STB layer, followed by infection of

Hofbauer cells (Figure 2a and b) or induced migration of infected maternal MACs towards the fetal placenta may be employed (Figure 2d)."

We have now rephrased the sentence including the right references:

"Damage of the STB layer, antibody-mediated transcytosis of CMV through STB, followed by infection of Hofbauer cells (Figure 3a and c) or induced migration of infected maternal MACs towards the fetal placenta may be involved (Figure 3b)."

4.2 (the CNS):

CNS barriers:

1st paragraph (page 16):

- Citation 100 does not fit here, a different citation is needed

We appreciate the thorough revision and apologize for the mistake. We have now referenced the correct work by

Patel N & Kirmi O (2009) Anatomy and Imaging of the Normal Meninges. *Seminars in Ultrasound, CT and MRI* 30: 559–564

2nd paragraph (page 16):

- The first citation (106) is not sufficient for all the previous information; another citation is needed (prob. 105).

Thank you, we have rephrased the sentence and added the additional citation.

"The formation of the BBB is a multi-step process that begins with sprouting angiogenesis directed by the neuroepithelium during later organogenesis".

Risau W (1997) Mechanisms of angiogenesis. *Nature* 386: 671–674

Abbott NJ, Patabendige AAK, Dolman DEM, Yusof SR & Begley DJ (2010) Structure and function of the blood–brain barrier. *Neurobiology of Disease* 37: 13–25

Figure 3:

- A: the connection of the CMV infection, shown directly adjacent to the letter a, to the resulting inflammatory signals that activate astrocytes that in turn lead to proteasomal degradation of the tight junctions is not clear. The four parts of the figure (CMV infection; microglia and astrocyte activation by inflammatory cytokines resulting from the CMV infection; production of additional cytokines + activation of MMP9; degradation of tight junctions) should be connected better to

show the step-by-step progression. Additionally, the tight junction disruption symbol should be changed to a different color, as it is nearly not visible on the red background.

Thank you for highlighting the need to adjust this figure. We have now designated a separate part in the figure to “IFN- γ \rightarrow microglia activation \rightarrow TNF production \rightarrow i) astrocyte activation; ii) neuronal MMP9 production” (a) CMV induced IFN- γ activates microglia, leading to secretion of TNF, which may disrupt astrocyte-endothelium communication or promote MMP9 production in neurons, thereby compromising tight junction integrity.

Also, we have refined the depiction of tight junction disruption.

- E: the part e is mentioned in the figure legend but is not shown in the figure itself or in the text. We have now restructured the figure and the references.

- BV (blood vessel), CSF (cerebrospinal fluid) and Ven (ventricular system) as abbreviations are missing; the abbreviations for Ep (ependymal cell layer) as well as CP (choroid plexus) are mentioned in the legend but do not occur in the figure. I would suggest to write out most of these abbreviations in the figure to make the figure more self-explanatory.

We thank the reviewer for this advice. We have now written out most of the structures and cells in the figure and only used as many abbreviations as necessary.

CNS-associated MACs:

2nd paragraph (page 20):

- "The most abundant immune cells in the dura are dural MACs that experience an influx of monocytes" - Which influx? Do the dural MACs result from this influx or is the influx of those monocytes additional to the already existing dMACs?

The influx of those monocytes results in monocyte-derived MHCII⁺ dMACs in addition to embryonic MHCII⁻ dMACs. We have now rephrased these sentences to make it more clear.

“dMACs share a common progenitor with microglia (Kierdorf *et al*, 2013; Goldmann *et al*, 2016).

In contrast to microglia, a subset of dMACs experience an active turnover by monocyte influx throughout life. These monocytes migrate via direct ossified connections from the cranial bone marrow to the dura mater in steady state (Cugurra *et al*, 2021; Herisson *et al*, 2018) giving rise to MHCII⁺ dMACs alongside embryonic MHCII⁻ dMACs (Goldmann *et al*, 2016; Van Hove *et al*, 2019).”

3rd paragraph (page 21):

- Missing s: "Stromal cpMACs are partially replenished by monocytes already in steady state [128]"

We have added the “s” to monocytes.

4th paragraph (page 21):

- "Although under dispute, mMACs may also access the bloodstream with their processes, especially during inflammation [135]" - What do the authors mean when writing about "processes"?

We apologize for the unclear term. We meant macrophage protrusion and corrected it accordingly.

“Although under dispute, mMACs may also access the bloodstream with their protrusions, especially during inflammation (Barkauskas *et al*, 2013).”

Referee #3:

Major issues:

1. MCMV infection of mice serves as a convenient and tractable model to study CMV pathogenesis in vivo. There are many similarities between mice and humans, and MCMV and HCMV. However, there are also differences, which should not be swept under the rug. In most parts of this review, the authors treat the two systems as equal and mix findings obtained in the mouse system with those obtained in human patients. In my opinion, this is not appropriate. Therefore, I suggest that the authors separate the findings from the two systems in each chapter. In many chapters, the results rely primarily on the MCMV/mouse system. This could be stated at the beginning of the chapter. Towards the end of the chapter, the authors could summarize the findings from the human system and discuss how they agree (or disagree) with the findings from the mouse system. In a few cases, the current state of knowledge is largely based on data from human studies (e.g., GI infections, encephalitis in adults). Here, the authors could turn it around and first report the findings from the human system before comparing them to those from the mouse system.

We strongly agree with the reviewer that, despite many similarities, important differences exist between the mouse/MCMV and human/HCMV systems. We also agree that careful distinction between experimental models is essential for accurate interpretation of the literature. However, for reasons of readability and narrative flow, we have chosen to maintain the current overall structure of the review. To address the reviewer's concern, we have now revised the manuscript to stringently and consistently specify the experimental system and CMV species (MCMV, HCMV, or other CMVs) throughout the text. In addition, we now clearly state at the beginning of each section which model system provides the majority of the discussed data (e.g., placenta—primarily human studies; lung and brain—predominantly mouse/MCMV studies). We believe that these revisions substantially improve clarity while preserving the coherence of the manuscript.

2. Myeloid cells, particularly DCs and MACs, play an important role in the adaptive immune response. Surprisingly, this aspect is barely covered in this review. It would be fair to devote at least one paragraph to the interaction between myeloid and lymphoid cells and how this interaction is modulated by HCMV and MCMV. The authors could direct the reader to other review articles if they feel that detailed information would go beyond the scope of the present review.

We thank the reviewer for this thoughtful idea. We realized that our (sub)title for this section was too broad and naturally would raise these expectations. However, we have now adjusted the title

to underline its defined scope: “The myeloid cell lineage and its nexus with CMV replication, dissemination and latency”.

Minor points:

3. The authors should mention in the abstract that results from both human and murine systems are being reviewed and discussed.

We have now added the following sentences to the abstract: “The review integrates evidence from multiple CMV species, with emphasis on human and mouse data.”

4. In some instances, general statements are not underpinned by appropriate citations. In the Introduction, for instance, reference #1 is not ideal to support the statements. More recent and more focused reviews are available. The same is true for reference #2. There are no references at all for the statements in the remainder of the paragraph.

We have now added more references to this paragraph to back up our statements.

5. Introduction, page 2. "microbial rich and immunologically demanding" -> please rephrase.

We have now rephrased the sentences: “Birth represents a profound physiological shift for the organism, including the abrupt transition from a protected intrauterine environment to one that is densely populated by microbes. This transition necessitates rapid adaptation of multiple organ systems, including the immune system (Henneke *et al*, 2021).”

6. Page 3. "manipulating structural and especially myeloid immune cells". Please define structural immune cells.

We apologize for the unclear wording. We initially meant structural cells and immune cells, because we will later also discuss endothelial cells, astrocytes, trophoblasts additionally to myeloid cells. However, during revision of language and grammar of our manuscript, as asked by Referee #1, we have now rephrased the sentence:” Hence, understanding the strategies that CMVs use to breach barriers, including activating, infecting and/or manipulating cells in barrier tissues is, both mechanistically and translationally, the key to decipher the establishment of CMV infection in its host.

7. Page 4. It is not true that M36 "specifically prevents programmed cell death in MAC", although it is crucial for viral replication in MAC in vitro and dissemination in vivo. However, there are other MCMV and HCMV cell death suppressors of similar importance for MAC infection, which were not mentioned.

We agree with the reviewer that our original text was somewhat understandable. So, we reformulated the text to express the MAC specific issues of M36 more specifically. M36, an inhibitor of caspase-8 in all infected host cells, induces a MAC specific growth phenotype *in vitro* and is required for MCMV dissemination *in vivo* (Ebermann *et al*, 2012). Its HCMV homolog UL36 also inhibits apoptosis in the human monocyte/MAC cell line THP1 and cloning of UL36 into M36-deleted MCMV rescues viral infection kinetics in murine MAC cell line RAW264.7 and *in vivo* (McCormick *et al*, 2010; Chaudhry *et al*, 2017).

8. Page 8. "Our previous study identified". One should generally not write review articles using the first person, as academic and formal writing emphasizes objectivity by avoiding "we" and "our". Instead, one should focus on the literature and findings, use the third-person or passive voice, and state conclusions directly without centering them on one's personal perspective.

We appreciate the reviewer's view on it and strongly agree. Therefore, we have rephrased the sentence: "Interestingly, depletion of alveolar MACs not only leads to less virus in the lung, but also to reduced viral dissemination to liver or brain in neonatal mice (Stahl *et al*, 2015). Infected alveolar MACs that were adoptively transferred from *Itgax*^{cre/+}:ROSA26^{LSL-Tomato} into *Csf2rb*^{-/-} mice translocate to the lung interstitium and migrate to the mediastinal lymph node after infection of adult mice (Baasch *et al*, 2021). These findings indicate a shared MCMV strategy that exploits the reprogramming of alveolar MACs into invasive, migratory cells to enable dissemination independent of age.

9. Figure 1. It is not helpful to coin new abbreviations that are not used in the literature. The commonly used abbreviation for the viral glycoprotein O is gO, not vgO, and the M33 protein is usually called M33 or pM33, but not vM33.

We initially wanted to clarify that these are viral gene products, but the reviewer is right, the correct nomenclature should be used, especially for readers that are very familiar with CMVs. We have changed the abbreviations accordingly.

10. Page 11. "Intragastral infection" -> gastric or intragastric (?)

We thank the reviewer for highlighting this mistake. We use now the right term "intragastric" infection via oral gavage.

11. Page 13. The "evidence supporting IgG-dependent transcytosis via FcRn across the STB layer" needs a better explanation. If FcRn facilitate transcytosis of viruses bound antibodies, why aren't all viruses with a viremic phase transmitted to the fetus? Does this imply that non-neutralizing HCMV-specific antibodies facilitate transmission across the STB layer?

The reviewer raises a compelling point that warrants further investigation. HCMV possesses a diverse array of glycoproteins embedded within its envelope. We hypothesize that CMV-encoded IgG-binding glycoproteins, which have been previously identified and are unique to CMVs, are prime candidates for this mechanism.

Dr. Sebastian Baasch
University Medical Center, Faculty of Medicine, University of Freiburg
Institute for Infection Prevention and Control
Breisacher Str. 115B
Freiburg 79106
Germany

Dear Dr. Baasch,

I am pleased to inform you that your review manuscript has been accepted for publication in EMBO reports. Your manuscript will be processed for publication by EMBO Press. It will be copy edited and you will receive page proofs prior to publication. Please note that you will be contacted by Springer Nature Author Services to complete licensing information.

Yours sincerely,

>>> Please note that it is EMBO Reports policy for the transcript of the editorial process (containing referee reports and your response letter) to be published as an online supplement to each paper. If you do NOT want this, you will need to inform the Editorial Office via email immediately. More information is available here: <https://link.springer.com/partners/embo-press/editorial-policies#Peer%20review>